# Arl2 GTPase associates with the centrosomal protein Cdk5rap2 to regulate cortical development via microtubule organization

Dongliang Ma[1]*, Kun-Yang Lin[1], Divya Suresh[1], Jiaen Lin[1], Mahekta R. Gujar[1], Htet Yamin Aung[1], Ye Sing Tan[1], Yang Gao[1], Anselm S. Vincent[1], Teng Chen[2,3], Hongyan Wang[1,4,5]*

1 Program in Neuroscience and Behavioural Disorders, Duke-NUS Medical School, Singapore, 2 College of Forensic Medicine, Xi'an Jiaotong University Health Science Center, Xi'an, Shaanxi, PR China, 3 The Key Laboratory of Health Ministry for Forensic Science, Xi'an Jiaotong University, Shaanxi, PR China, 4 Department of Physiology, Yong Loo Lin School of Medicine, National University of Singapore, Singapore, 5 Integrative Sciences and Engineering Programme, National University of Singapore, Singapore

* dongliang.ma@duke-nus.edu.sg (DM); hongyan.wang@duke-nus.edu.sg (HW)

**Data Availability Statement:** All relevant data are within the paper and its Supporting Information files.

## Abstract

ADP ribosylation factor-like GTPase 2 (Arl2) is crucial for controlling mitochondrial fusion and microtubule assembly in various organisms. Arl2 regulates the asymmetric division of neural stem cells in *Drosophila* via microtubule growth. However, the function of mammalian Arl2 during cortical development was unknown. Here, we demonstrate that mouse Arl2 plays a new role in corticogenesis via regulating microtubule growth, but not mitochondria functions. Arl2 knockdown (KD) leads to impaired proliferation of neural progenitor cells (NPCs) and neuronal migration. Arl2 KD in mouse NPCs significantly diminishes centrosomal microtubule growth and delocalization of centrosomal proteins Cdk5rap2 and γ-tubulin. Moreover, Arl2 physically associates with Cdk5rap2 by in silico prediction using AlphaFold multimer, which was validated by co-immunoprecipitation and proximity ligation assay. Remarkably, Cdk5rap2 overexpression significantly rescues the neurogenesis defects caused by Arl2 KD. Therefore, Arl2 plays an important role in mouse cortical development through microtubule growth via the centrosomal protein Cdk5rap2.

## Introduction

Neural stem cells (NSCs) play a central role in the development of the mammalian brain. Cortical NSCs reside in the ventricular zone (VZ) and subventricular zone (SVZ), namely, neuroepithelial cells and the apical radial glial cells, self-renew, and proliferate to generate neurons that migrate to the cortical plate (CP) [1–8]. Both types of cells are collectively termed as neural stem and progenitor cells, hence referred to as neural progenitor cells (NPCs). NPCs divide either symmetrically or asymmetrically [1,2]. Symmetric division of NPCs expands the stem cell pool during early neurogenesis [1,2]. Subsequently, NPCs divide asymmetrically to generate intermediate progenitor cells that divide once to produce 2 neurons [1,9]. The balance between the proliferation and differentiation of NPCs has a direct impact on neuron

**Funding:** This work is supported by the Singapore Ministry of Education Tier 2 MOE-T2EP30121-0002 to H.W.. The funders had no role in study design, data collection and analysis, decision to publish, or preparation of the manuscript.

**Competing interests:** The authors have declared that no competing interests exist.

**Abbreviations:** Arl2, ADP ribosylation factor-like GTPase 2; BSA, bovine serum albumin; Cdk5rap2, CDK5 Regulatory Subunit Associated Protein 2; CP, cortical plate; DCX, doublecortin; EdU, 5-ethynyl-2′-deoxyuridine; GFP, green fluorescent protein; ipTM, interface pTM; IUE, in utero electroporation; IZ, intermediate zone; KD, knockdown; mNPC, mouse neural progenitor cell; MTOC, microtubule-organizing center; NPC, neural progenitor cell; NSC, neural stem cell; PBS, phosphate buffered saline; PCM, pericentriolar material; PLA, proximity ligation assay; SDC-SIM, spinning disk confocal-structured illumination microscopy; shRNA, short hairpin RNA; SVZ, subventricular zone; TBCD, Tubulin folding cofactor D; VZ, ventricular zone; WB, western blot.

formation. Moreover, defects in NPC proliferation are associated with neurodevelopmental disorders [10–12].

Centrosomal proteins play crucial roles during mouse cortical development. The centrosome, composed of a pair of centrioles surrounded by pericentriolar material (PCM) protein, is the major microtubule-organizing center (MTOC) that contributes to the formation of the mitotic spindle during cell division. A few centrosomal proteins including PCM1 and Cep120 play critical roles in brain development and variants in these 2 genes are associated with primary microcephaly [13–15]. CDK5 Regulatory Subunit Associated Protein 2 (Cdk5rap2/ Cep215) is an evolutionarily conserved PCM protein that plays a crucial role in centrosomal duplication and maturation as well as microtubule organization in various organisms. Cdk5rap2 is critical for proliferation and differentiation of neuronal progenitor cells during mouse cortical development [16,17]. Mutations in Cdk5rap2 are associated with congenital diseases such as primary microcephaly and primordial dwarfism [18,19].

Arl2 (ADP-ribosylation factor-like 2) is an evolutionarily conserved small GTPase that is crucial for the formation of microtubules and maintaining centrosome integrity [20,21]. Arl2 cycles between an inactive GDP-bound and an activated GTP-bound state and is a regulator of tubulin folding and microtubule biogenesis [9,20–24]. Yeast orthologue of Arl2, together with TBCD and TBCE, forms a tubulin chaperone for microtubule biogenesis [25]. We previously showed that *Drosophila* Arl2 is essential for NSC polarity and microtubule growth [22]. Arl2 also plays a role in mitochondrial dynamics and function [24]. Arl2 regulates mitochondrial fusion when it is in the intermembrane space [26]. Arl2 also interacts with mitochondrial outer membrane proteins Miro1 and Miro2 to modulate mitochondrial transport and distribution [27]. Variants in human ARL2 and ARL2BP have been identified in eye disorders, namely, MRCS (microcornea, rod-cone dystrophy, cataract, and posterior staphyloma) syndrome and retinitis pigmentosa, respectively [28,29].

Mammalian Arl2 is widely expressed in various tissues and is most abundant in the brain [30]. However, the role of mammalian Arl2 during brain development has not been established. In this study, we demonstrate a novel role for the mammalian Arl2 in cortical development. We show that Arl2 is required for the proliferation, migration, and differentiation of mouse forebrain NPCs in vitro and in vivo by regulating centrosome assembly and microtubule growth in NPCs. Moreover, Arl2 colocalizes with Cdk5rap2 at the centrosomes and can physically associate with it. Finally, Arl2 functions upstream of Cdk5rap2 in regulating NPC proliferation and migration during mouse cortical development.

## Results

### Arl2 knockdown results in a reduction in mNPC proliferation and neuronal migration

To understand the functions of Arl2 in the developing cortex, we first examined the expression pattern of Arl2 in mouse brains. We analyzed Arl2 protein levels by western blot (WB) in mouse cerebral cortical tissue isolated from E12, 13, 14, 16, P1, P7, respectively. We find that Arl2 protein levels are highest at embryonic stages (E12, 13, 14, 16) as compared to the postnatal stages (P1, P7) (S1A and S1B Fig). Reanalyzing a published scRNA-seq dataset, which profiled the embryonic mouse brain [31], suggests that Arl2 exhibits the highest expression in radial glial cells, followed by neurons, but shows low expression levels in glial cells and other cell types (S1C Fig). This further supports the role of Arl2 during the mouse cortical development.

To examine the role of mouse Arl2 in cortical development, we silenced endogenous Arl2 expression in the primary culture of mouse neural progenitor cells (mNPCs) isolated from E14 mouse cortex in vitro (S1D Fig) as well as in utero electroporated cells in mouse brain in vivo

(Fig 1A) using short hairpin RNA (shRNA). We identified 2 independent shArl2-1 and shArl2-2 tagged with green fluorescent protein (GFP), which are capable of knocking down endogenous Arl2 expression. Upon Arl2 knockdown (KD), Arl2 protein detected by anti-Arl2 antibodies in the WB was reduced to 42.53% (shArl2-1) and 20.38% (shArl2-2), respectively, compared with the control. Since KD by shArl2-2 is more efficient (S1E and S1F Fig), we used shArl2-2 for the subsequent phenotypic analysis.

To determine whether Arl2 KD affects the proliferation and differentiation of mNPCs during mouse cortical development, we silenced endogenous Arl2 expression in the primary culture of mNPCs in vitro by lentivirus (pPurGreen) infection in 48-h culture and pulse-labelled with 5-ethynyl-2′-deoxyuridine (EdU) for 3 h before harvesting the cells. Remarkably, we observed a significant decrease in the proportion of EdU+ cells to 28.78% in the shArl2 group as compared to the control group (56.25%) (S1G and S1H Fig). We also observed that there was a significant increase in the proportion of Caspase-3$^+$ GFP$^+$ cells in the Arl2 KD mNPCs (25.86 ± 3.26%) as compared to control (9.64 ± 0.86%) (S1G and S1I Fig).

Consistent with these observations, there was a significant reduction in cell proliferation upon Arl2 KD in mNPCs as compared to control in automated live-imaging analysis by Incucyte (S1J and S1K Fig and S1 and S2 Movies). Taken together, our data suggests that Arl2 is required for the proliferation of mNPCs in vitro.

To further assess the impact of Arl2 KD on the proliferation, differentiation, and migration of mNPCs during cortical development in vivo, we introduced Arl2 shRNA-2 via microinjection into the lateral ventricle of mouse embryos, followed by in utero electroporation (IUE) at embryonic day 13 (E13) (Fig 1A), and examined cortical neurogenesis from E14 to E16. At E14, 1 day after IUE and following 6-h pulse-labelling with EdU before sample collection, the majority of GFP-positive cells were located in the VZ and SVZ, with the minority population of cells migrating into the intermediate zone (IZ) in both control and shArl2 groups (Fig 1B). Interestingly, in the VZ and SVZ, we observed a substantial reduction in the proportion of EdU+/GFP+ double-labelled cells in the shArl2 group (42.43%) as compared to the control group (58.25%) (Fig 1C). These data strongly suggest a notable impairment in the proliferation of NSCs due to Arl2 KD in vivo. At E15, majority of control GFP-positive (GFP+) cells were located in the IZ (49.99%) or had migrated into the CP (12.68%), with a few GFP+ remaining in the VZ (19.92%) and SVZ (18.76%) (TBR2 labels the SVZ and Pax6 labels the VZ; Fig 1D and 1E). In contrast, Arl2 KD resulted in significantly more cells remaining in the VZ (30.94%), SVZ (27.64%), and IZ (38.00%), with fewer cells at the CP (3.42%) as compared to the control group (Fig 1D and 1E). At E16, 3 days after IUE, more GFP+ cells were located in the IZ (52.82%) and CP (demarcated by TBR1; 23.07%), while the rest of the cells persisted in the VZ (10.17%) and SVZ (13.94%) in the control group (Fig 1F and 1G). Remarkably, Arl2 KD caused a notable retention of cells in the VZ (21.42%), SVZ (18.82%), and IZ (48.00%), accompanied by much fewer cells at the CP (11.75%) (Fig 1F and 1G), suggesting defects in neuronal migration and/or neuronal differentiation.

## Loss of Arl2 results in defects in neuronal differentiation of mNPCs

To investigate whether Arl2 regulates neuronal differentiation of mNPCs, we examined NeuroD2, a neuronal marker found in newly born neurons in IZ and immature/mature neurons in CP and quantified the proportion of NeuroD2+GFP+ cells in IZ 3 days after IUE. At E16, 3 days after IUE, we observed a significant decrease in the proportion of NeuroD2+GFP+ cells in the shArl2 group (42.06 ± 4.42%) as compared to the control group (75.49 ± 3.77%; Fig 2A and 2B). These data suggest a notable impairment in the differentiation of NSCs due to Arl2 KD in vivo.

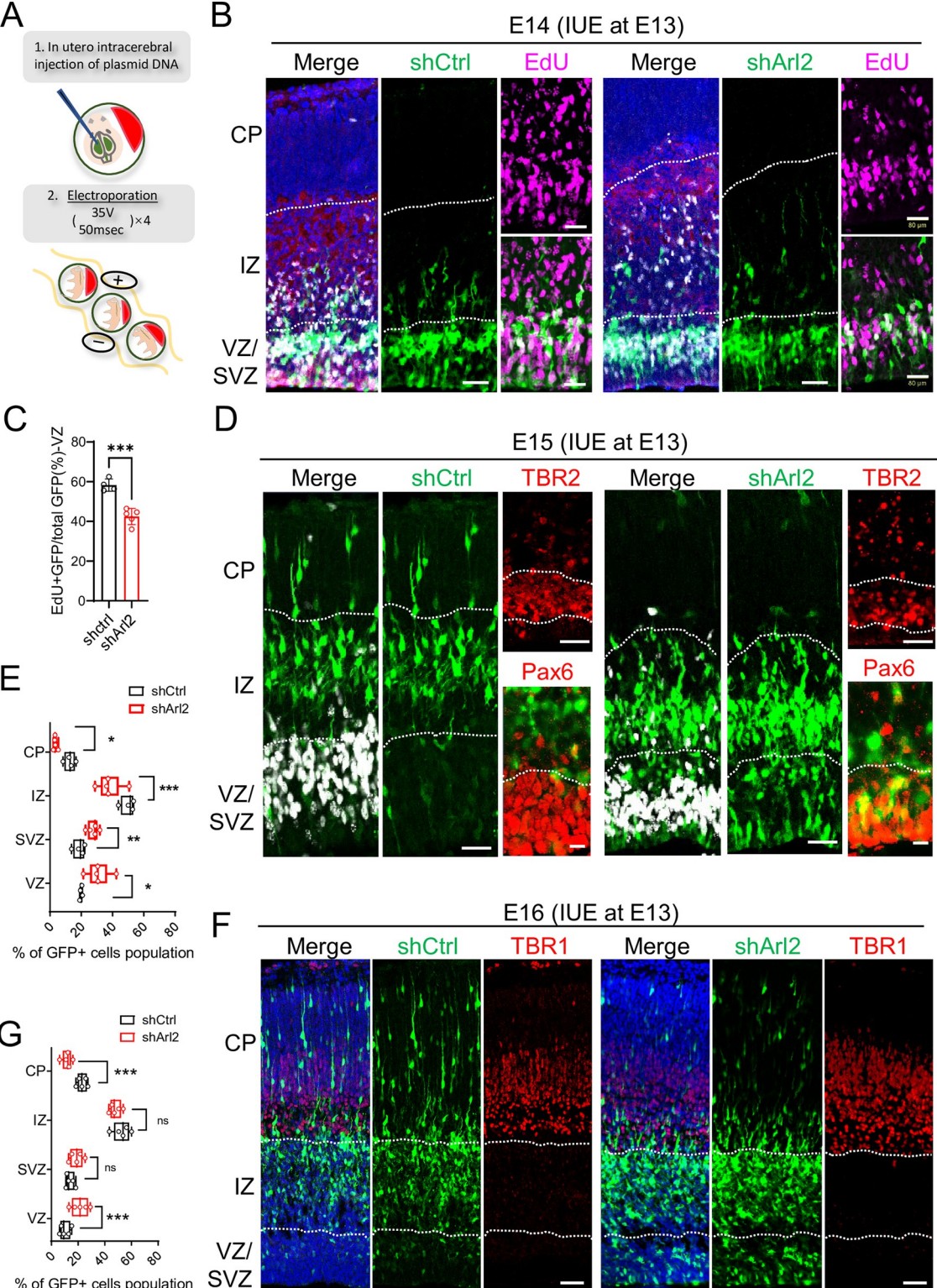

**Fig 1. Arl2 KD results in a reduction in mNPC proliferation and neuronal migration.** (**A**) Schematic representation of IUE. (**B**) Brain slices from shCtrl (scrambled control) and shArl2 (Arl2 shRNA) groups at E14, 1 day after IUE, were labelled with EdU and GFP. (**C**) Bar graph showing reduced EdU incorporation upon Arl2 KD (42.43 ± 4.04% in shArl2 vs. 58.25 ± 3.11% in shCtrl). The values represent the mean ± SD (shCtrl: *n* = 4 embryos, shArl2: *n* = 5 embryos). Student *t* test, differences were considered significant at ***$p < 0.001$. (**D**) Cortical brain sections from shCtrl and shArl2 groups at E15, 2 days after IUE, were labelled with

TBR2 (intermediate neural progenitor marker and labelling SVZ) or Pax6 (radial glia marker and labelling VZ) with GFP. (**E**) Box plots representing GFP+ cells in CP (shCtrl: 12.68 ± 3.47%, shArl2: 3.42 ± 1.17%), IZ (shCtrl: 49.99 ± 4.42%, shArl2: 38.00 ± 7.91%), SVZ (shCtrl: 18.76 ± 3.71%, shArl2: 27.64 ± 3.53%), and VZ (shCtrl: 19.92 ± 0.87%, shArl2: 30.94 ± 7.51%) showing defects in neuronal migration to CP upon Arl2 KD compared to the control. The values represent the mean ± SD (shCtrl, $n = 4$ embryos; shArl2, $n = 5$ embryos). Multiple unpaired $t$ tests, differences were considered significant at $*p < 0.05$, $**p < 0.01$, and $***p < 0.001$. (**F**) Cortical brain sections for shCtrl and shArl2 groups at E16, 3 days after IUE, were immunolabelled with TBR1 (immature neuron marker and labelling CP) and GFP. (**G**) Box plots representing GFP+ cells in CP (shCtrl: 23.07 ± 3.61%, shArl2: 11.75 ± 3.67%), IZ (shCtrl: 52.82 ± 6.31%, shArl2: 48.00 ± 4.24%), SVZ (shCtrl: 13.94 ± 3.15%, shArl2: 18.82 ± 4.90%), and VZ (shCtrl: 10.17 ± 3.85%, shArl2: 21.42 ± 6.38%), showing defects in neuronal migration to CP upon Arl2 KD compared to the control. Multiple unpaired $t$ tests, differences were considered significant at $***p < 0.001$. ns = nonsignificance. Scale bars; B and D = 100 μm, F = 150 μm. Source data can be found in S1 Data. Arl2, ADP ribosylation factor-like GTPase 2; CP, cortical plate; EdU, 5-ethynyl-2′-deoxyuridine; GFP, green fluorescent protein; IUE, in utero electroporation; IZ, intermediate zone; KD, knockdown; mNPC, mouse neural progenitor cell; SVZ, subventricular zone; VZ, ventricular zone.

Next, we examined doublecortin (DCX), a neuronal marker that is expressed in neuronal precursor cells and immature neurons. Interestingly, Arl2 KD in mNPCs led to a significant reduction in the population of DCX-positive cells (31.48 ± 1.17%) as compare to control (40.63 ± 4.11%; Fig 2C and 2D). Moreover, the average neurite length (52.02 ± 11.08 μm) as well as average neurite number (3.17 ± 0.47) were significantly reduced in Arl2 KD as compared to control (neurite length = 88.77 ± 4.69 μm, neurite number = 5.22 ± 0.14; Fig 2C, 2E and 2F). These observations further support the notion that loss of Arl2 affects neuronal differentiation of mNPCs.

## Arl2 regulates neuronal migration and neuritogenesis of neurons

To determine whether defective migration of VZ/SVZ cells is caused by neuronal migration and/or defective differentiation from progenitors to neurons, we examined NeuroD2, a neuronal marker, and quantified the migration distance of NeuroD2+GFP+ cells in the CP 4 days after IUE. We find that KD Arl2 resulted in significantly less migration of NeuroD2+GFP + cells (104.3 ± 13.05 μm) as compared to the control group (135.8 ± 9.81 μm; Fig 2G and 2H). This observation supports neuronal migration defect upon Arl2 KD in vivo. Consistent with these in vivo findings, Arl2 KD in mouse primary neurons in vitro dramatically affected neuronal morphology, as fewer and shorter neurites were observed in these neurons as compared to control (S2A–S2D Fig). The total intersection number as measured by Sholl's analysis was significantly reduced in Arl2 KD (22.78 ± 5.04) as compared to control (64.17 ± 2.62; S2A–S2D Fig). These data suggest that neuronal migration defects caused by Arl2 dysfunction is possibly due to its role in regulating neurite outgrowth.

In summary, our findings collectively highlight the critical role of Arl2 in the proliferation, differentiation of NSCs, and the migration of neuronal cells during cortical development.

## Overexpression mArl2 or hArl2 results in an increase in neuronal migration

Since the Arl2 KD led to decreased proliferation and migration of neuronal cells, we wondered whether overexpressing Arl2 had any effect on cortical development. We overexpressed both the human and mouse forms of wild-type Arl2 (FUtdTW-Arl2$^{WT}$) driven by UbC promotor via microinjection into the lateral ventricle of mouse embryos, followed by IUE at E13 (Fig 3A and 3B). Interestingly, at E16, 3 days after IUE, there was a significant increase in neuronal cell migration (tdTomato-positive (Td+) cells) to the CP in human Arl2$^{WT}$ (hArl2$^{WT}$) (VZ = 11.13%, SVZ = 18.21%, IZ = 40.25%, CP = 30.41%) as compared to control (VZ = 13.23%, SVZ = 19.47%, IZ = 45.92%, CP = 21.39%). Likewise, overexpression of mouse Arl2$^{WT}$ (mArl2$^{WT}$) also resulted in a significant increase in neuronal migration (Fig 3A and 3B, VZ = 13.40%, SVZ = 19.13%, IZ = 36.60%, CP = 30.86%) as compared to control,

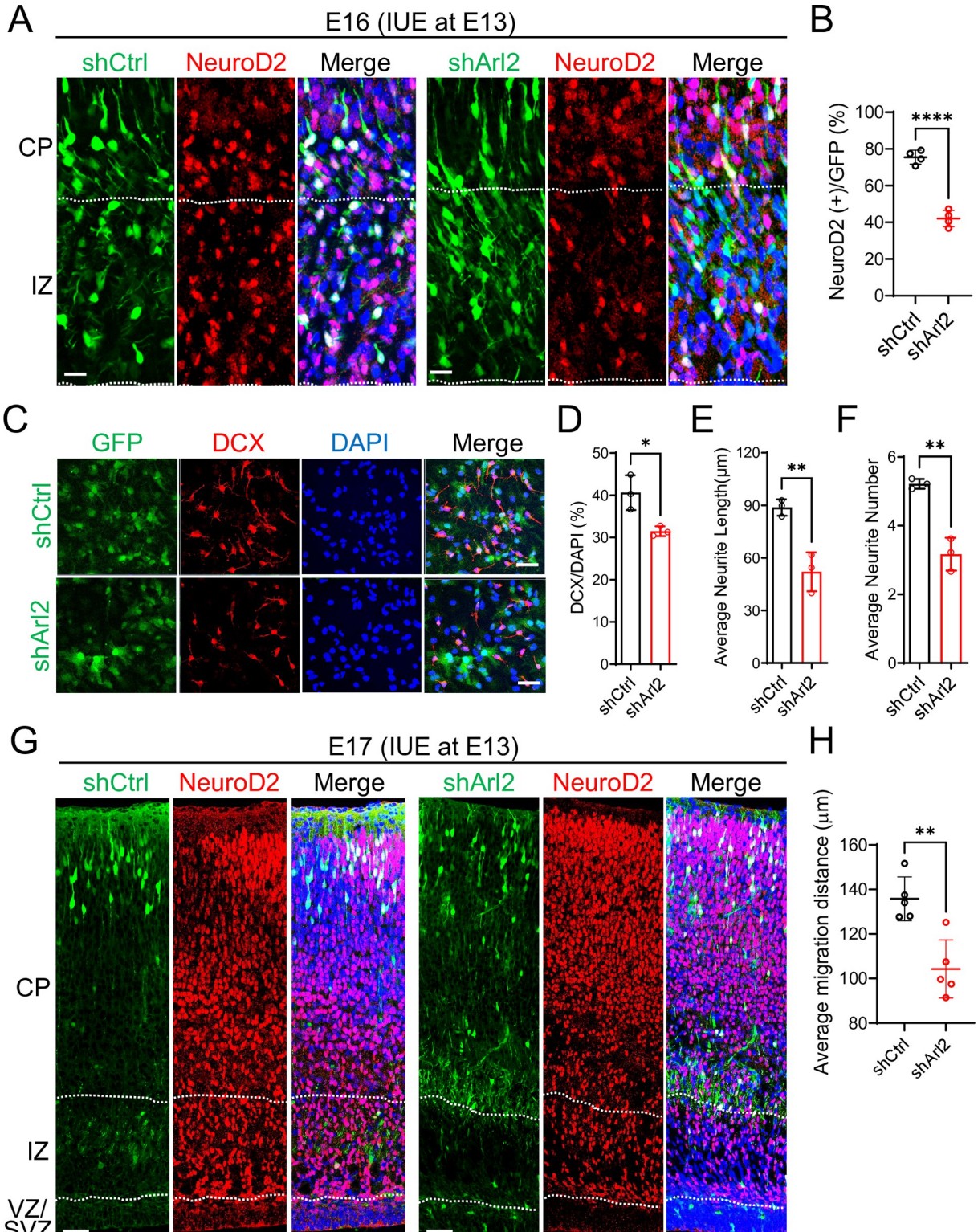

**Fig 2. Loss of Arl2 led to defects in mNPC differentiation and neuronal migration.** (**A**) Cortical brain sections for shCtrl and shArl2 groups at E16, 3 days after IUE, were immunolabelled with NeuroD2 (neuronal marker found in newly born neurons in IZ and immature/mature neurons in CP) and GFP. (**B**) Box plots representing the proportion of NeuroD2+GFP+ cells in IZ 3 days after IUE (shCtrl: 75.49 ± 3.77%, shArl2: 42.06 ± 4.42%). (**C**) Immunostaining micrographs showing the DCX+ immature neurons after 5 days mNPC differentiation assay in both shCtrl and shArl2 groups. (**D-F**) Bar graphs representing the population of DCX-positive cells (shCtrl: 40.63 ± 4.11%, shArl2:

31.48 ± 1.72%); the average neurite length (shCtrl: 88.77 ± 4.69 μm, shArl2: 52.02 ± 11.08 μm) and the average neurite number (shCtrl: 5.22 ± 0.14, shArl2: 3.17 ± 0.48). The values represent the mean ± SD. Student $t$ test in D, E, and F, $n$ = 3. Differences were considered significant at $^*p < 0.05$, $^{**}p < 0.01$. (**G**) Cortical brain sections for shCtrl and shArl2 groups at E17, 4 days after IUE, were immunolabelled with NeuroD2 and GFP. (**H**) Box plots representing average migration of cells (shCtrl: 135.8 ± 9.81 μm, shArl2: 104.3 ± 13.05 μm), showing defects in neuronal migration to CP upon Arl2 KD compared to the control. We measured the migration distance of each neuron to the pia surface of the cortex using Image J, and the distance of each neuron is then normalized to the cortex thickness to obtain the average distance. The values represent the mean ± SD (shCtrl, $n$ = 5 embryos; shArl2, $n$ = 5 embryos). Student $t$ test in H. Differences were considered significant at $^{**}p < 0.01$. Scale bars; A = 25 μm, C = 50 μm, G = 150 μm. Source data can be found in S1 Data. Arl2, ADP ribosylation factor-like GTPase 2; CP, cortical plate; DCX, doublecortin; GFP, green fluorescent protein; IUE, in utero electroporation; IZ, intermediate zone; KD, knockdown; mNPC, mouse neural progenitor cell.

suggesting human and mouse forms Arl2 have conserved functions in neuronal migration (Fig 3A and 3B). Since human and mouse Arl2 show 96% homology and our overexpression results show similar phenotypes for both species, we used the human Arl2 for all subsequent overexpression experiments unless otherwise stated.

## Overexpression of mutant forms of Arl2 resulted in a defect in neuronal migration and neuritogenesis

To further elucidate the role of Arl2 in neurogenesis, we tested the effect of overexpression of wild-type Arl2 (Arl2$^{WT}$) as well as 2 mutant forms of Arl2, namely, the dominant-negative form (Arl2$^{T30N}$) and the dominant-active form (Arl2$^{Q70L}$). Expression of these constructs tagged with Tdtomato (FUtdTW) were driven by the Ubiquitin C (UbC) promotor, in the primary culture of mNPCs in vitro by lentivirus (FUtdTW) infection in 48-h culture and pulse-labelled with EdU for 3 h before harvesting the cells. Interestingly, there was a significant increase in the proportion of EdU+ cells in Arl2$^{WT}$ (79.01%) as compared to control (59.06%) (S3A and S3B Fig). In contrast, we observed a significant decrease in the proportion of EdU + cells in Arl2$^{T30N}$ and Arl2$^{Q70L}$ (36.4% and 27.4%, respectively) as compared to the control group (59.06%) (S3A and S3B Fig). Furthermore, overexpression of Arl2$^{T30N}$ and Arl2$^{Q70L}$ but not Arl2$^{WT}$ caused significant cell death as seen by the increase in caspase-3 staining in mNPCs in vitro as compared to control (S3A and S3C Fig; Control = 22.18%; Arl2$^{WT}$ = 30.89%; Arl2$^{T30N}$ = 86.76% and Arl2$^{Q70L}$ = 71.86%). Taken together, our data suggest that Arl2 is required for the proliferation of NSCs in vitro.

To assess the impact of Arl2 overexpression on the proliferation, differentiation, and migration of mNPCs during in vivo cortical development, we introduced Arl2 and 2 mutant forms via microinjection into the lateral ventricle of mouse embryos, followed by IUE at E13. Remarkably, at E16, overexpression of both mutant forms displayed a significant reduction in Td+ cells migrating to the cortical plate (Fig 3C and 3D; Arl2$^{T30N}$, VZ = 15.23%, SVZ = 19.34%, IZ = 65.43%, CP = 0.00% and Arl2$^{Q70L}$, VZ = 15.68%, SVZ = 20.80%, IZ = 63.53%, CP = 0.00%) as compared to control (Fig 3C and 3D; VZ = 14.23%, SVZ = 20.15%, IZ = 39.63%, CP = 25.99%). Our results suggest that overexpression of both Arl2$^{T30N}$ and Arl2$^{Q70L}$ results in a similar phenotype in neuronal migration as Arl2 KD.

Surprisingly, we find that overexpression of Arl2$^{T30N}$ and Arl2$^{Q70L}$ but not Arl2$^{WT}$ at E15, 2 days after IUE, showed a significant increase in the proportion of phospho-histone H3-positive (PH3+) cells in the VZ as compared to control (S3D and S3E Fig; control 3.31%, Arl2$^{WT}$ 2.94%, Arl2$^{T30N}$ 8.33%, Arl2$^{Q70L}$ 9.43%). This increase in PH3+ cells may be due to mitotic defects or overproliferation of radial glial cells. To distinguish these 2 possibilities, we examined the spindle poles marked by gamma-tubulin in these cells to assess their cell divisions. Indeed, overexpression of Arl2$^{T30N}$ and Arl2$^{Q70L}$ caused a significant increase in mitotic defects (cell cycle arrest), with defective spindle formation as compared to control and Arl2$^{WT}$ (S4A Fig). Furthermore, similar to our in vitro results, there was a significant increase in the

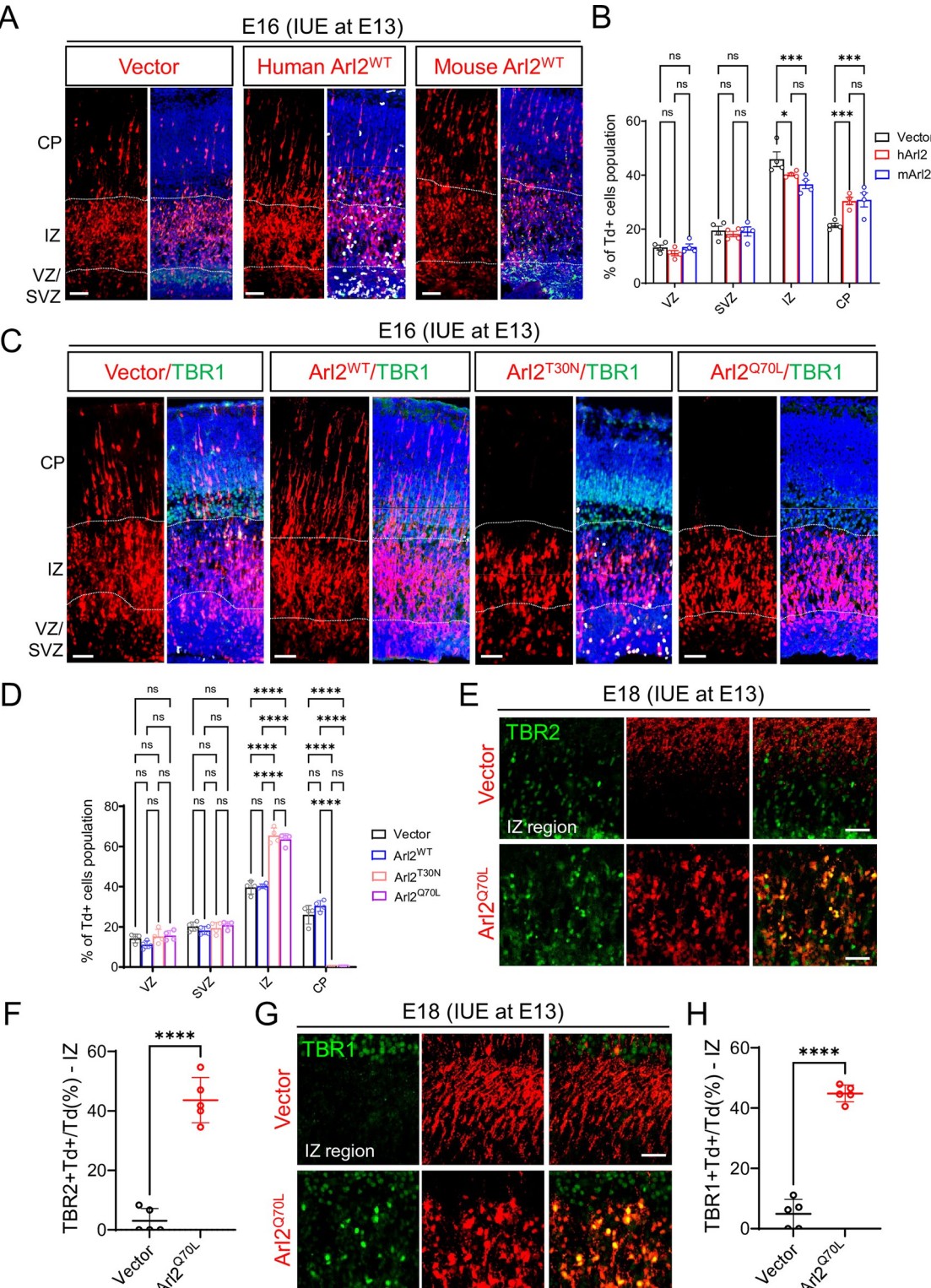

**Fig 3. Overexpression various forms of Arl2 alters neuronal migration.** (**A**) Cortical brain sections following overexpression of human Arl2$^{WT}$ (hArl2$^{WT}$), mouse Arl2$^{WT}$ (mArl2$^{WT}$) at E16, 3 days after IUE, were labelled with tdTomato (Td). (**B**) Bar graphs (images in A) representing Td+ cell population in the group of control (VZ = 13.23 ± 1.85%, SVZ = 19.47 ± 3.19%, IZ = 45.92 ± 5.44%, CP = 21.39 ± 1.58%, $n$ = 4 embryos), human Arl2$^{WT}$ (hArl2$^{WT}$) (VZ = 11.13 ± 1.90%, SVZ = 18.21 ± 1.98%, IZ = 40.25 ± 1.14%, CP = 30.41 ± 2.66%, $n$ = 4 embryos), mouse Arl2$^{WT}$ (mArl2$^{WT}$) (VZ = 13.40 ± 2.25%, SVZ = 19.13 ± 3.40%,

IZ = 36.61 ± 3.18%, CP = 30.86 ± 5.18%, $n$ = 4 embryos). (**C**) Cortical brain sections following overexpression of Arl2$^{WT}$, the dominant-negative form (Arl2$^{T30N}$) and the dominant-active form (Arl2$^{Q70L}$) at E16, 3 days after IUE, were immunolabelled with tdTomato (Td) and TBR1 (immature neuron marker and labelling CP). (**D**) Bar graphs (images in C) representing Td+ cell population in the group of control (VZ = 14.23 ± 2.15%, SVZ = 20.15 ± 2.25%, IZ = 39.63 ± 3.28%, CP = 25.99 ± 4.64%, $n$ = 4 embryos), Arl2$^{WT}$ (VZ = 11.13 ± 1.90%, SVZ = 18.21 ± 1.98%, IZ = 40.25 ± 1.14%, CP = 30.41 ± 2.66%, $n$ = 4 embryos), dominant-negative form (Arl2$^{T30N}$) (VZ = 15.23 ± 3.51%, SVZ = 19.34 ± 2.89%, IZ = 65.43 ± 3.78%, CP = 0.00 ± 0.00%, $n$ = 4 embryos) and dominant-active form (Arl2$^{Q70L}$) (VZ = 15.68 ± 2.44%, SVZ = 20.80 ± 1.84%, IZ = 63.53 ± 2.86%, CP = 0.00 ± 0.00%, $n$ = 4 embryos). (**E**) Cortical brain sections following overexpression of the dominant-active form (Arl2$^{Q70L}$) at E18, 5 days after IUE, were immunolabelled with tdTomato (Td), and TBR2 is intermediate progenitor cells marker. (**F**) Quantification graphs representing TBR2+Td+ cells population in the group of control (3.00 ± 4.15%) and Arl2$^{Q70L}$ mutants (43.61 ± 7.62%) in the IZ. (**G**) Cortical brain sections following overexpression of the dominant-active form (Arl2$^{Q70L}$) at E18, 5 days after IUE, were immunolabelled with tdTomato (Td), and TBR1 is immature neuron marker. (**H**) Quantification graphs representing TBR1+Td + cells population in the group of control (4.90 ± 4.83%) and Arl2$^{Q70L}$ mutants (44.84 ± 2.78%) in the IZ. The values represent the mean ± SD. Two-way ANOVA with multiple comparisons in C and D; Student $t$ test in F and G, $n$ = 5 embryos. Differences were considered significant at $^*p < 0.05$, $^{**}p < 0.01$, $^{***}p < 0.001$, and $^{****}p < 0.0001$, ns = nonsignificance. Scale bars; A and C = 150 μm, E and G = 100 μm. Source data can be found in S1 Data. Arl2, ADP ribosylation factor-like GTPase 2; CP, cortical plate; IUE, in utero electroporation; IZ, intermediate zone; SVZ, subventricular zone; VZ, ventricular zone.

proportion of caspase-3+ cells in the IZ in Arl2$^{Q70L}$ (38.52%) as compared to control (5.92%; S4B and S4C Fig). These data suggest that the migration and proliferation defects observed in Arl2 mutants are possibly due to mitotic defects, eventually leading to cell death.

To further test whether Arl2 affects neuronal migration, we examined TBR2, a transcription factor that marks the transition from radial glial cells to intermediate progenitor cells (Fig 3E and 3F). At E18, 5 days after IUE, we find that majority of the TBR2+Td+ cells have already migrated to the CP in control mouse brains, with few TBR2+Td+ cells still present in the IZ (Fig 3E and 3F). Interestingly, in Arl2$^{Q70L}$ mutants, a large population of TBR2+Td+ cells still remained in the IZ (Fig 3F, 43.61 ± 7.62%) with few to no cells migrating to the CP as compared to control (3.00 ± 4.15%). Similarly, a vast majority of neuronal (TBR1+) were still retained in the IZ in Arl2$^{Q70L}$ mutant mouse brains (Fig 3G and 3H; 44.84 ± 2.78%) as compared to control (4.90 ± 4.83%). In contrast, the expression of NeuroD2, a neuronal marker found in immature neurons, is significantly increased in Arl2$^{WT}$ (30.77 ± 2.93%) but dramatically reduced in Arl2$^{Q70L}$ (6.75 ± 2.69%) 3 days after IUE as compared to control (20.10 ± 3.11%) (S4D and S4E Fig).

To further determine whether neuronal migration defects caused by Arl2 dysfunction is due to impaired neuritogenesis, we examined the effect of Arl2$^{T30N}$ and Arl2$^{Q70L}$ in mouse primary neurons in vitro. Interestingly, overexpression of Arl2$^{T30N}$ and Arl2$^{Q70L}$ dramatically affected neuronal morphology, as fewer and shorter neurites were observed in these neurons as compared to control (S5A Fig). The total intersection number as measured by Sholl's analysis was significantly reduced in Arl2$^{Q70L}$ (11 ± 1.80) and Arl2$^{T30N}$ (8.83 ± 1.04) as compared to control (38 ± 4.58; S5A–S5C Fig). These data suggest that neuronal migration defects caused by Arl2 dysfunction is possibly due to its role in regulating neurite outgrowth.

Taken together, Arl2 dysfunction resulted in defective proliferation, differentiation of NSCs, and migration of neuronal cells.

## Arl2 dysfunction results in defects in cell cycle progression in mNPCs

To further understand how Arl2 affects mNPC proliferation, we performed live imaging of mNPCs in vitro using the Viafluor-488 live cell microtubule staining kit (Biotium, #70062) to look at cell cycle progression. In control mNPCs, the average time taken for a single mitotic cycle was 1.32 ± 0.21 h (Fig 4A and 4B and S3 Movie). Interestingly, Arl2 KD in mNPCs led to a significant increase in the mitotic duration (shArl2 = 6.55 ± 1.52 h) as compared to the control (Fig 4A and 4B and S3 Movie), suggesting an additional defect in the cell cycle progression caused by Arl2 KD. Similarly, mNPCs expressing Arl2$^{T30N}$ but not Arl2$^{Q70L}$ or Arl2-WT

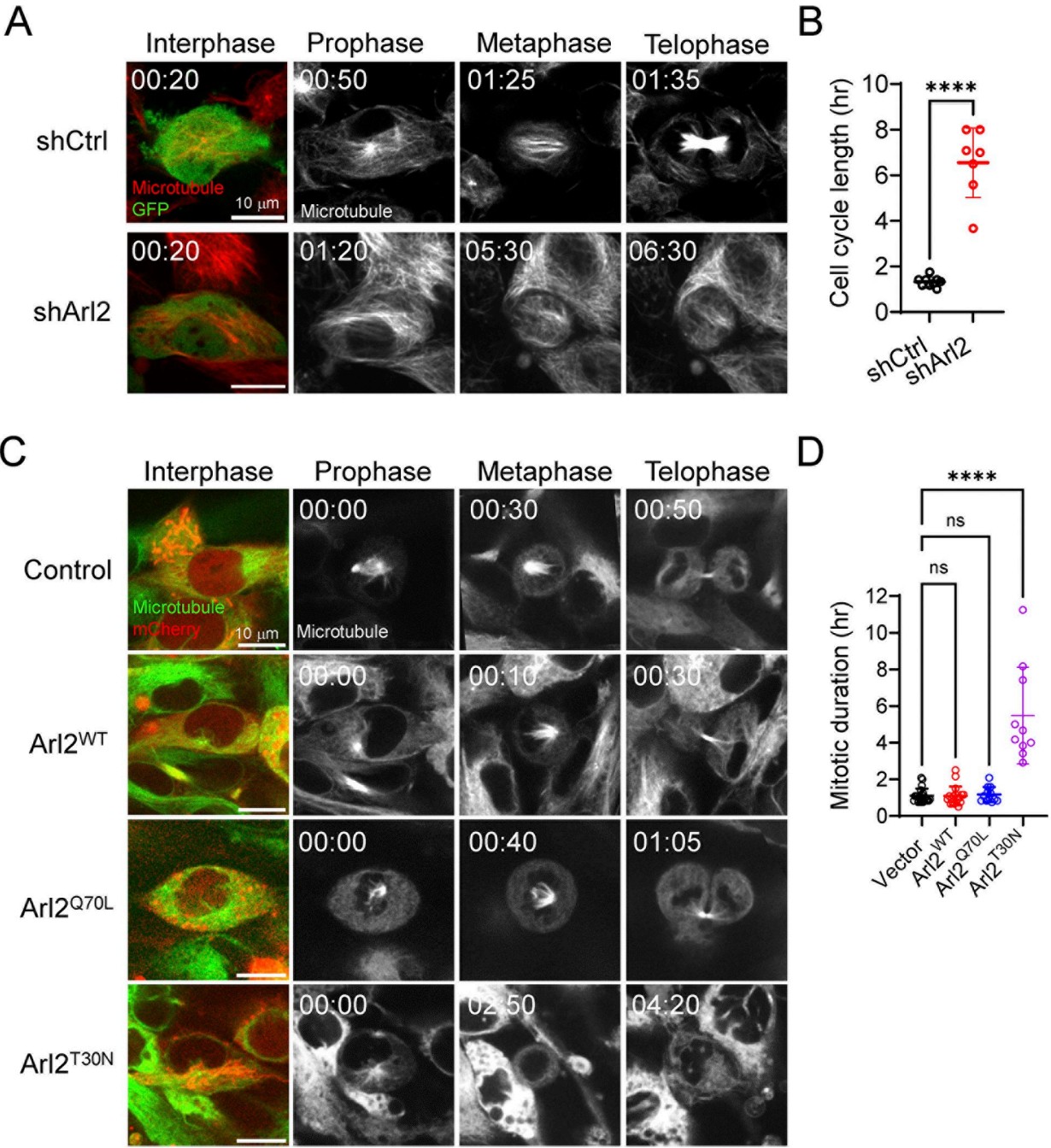

**Fig 4. Arl2 dysfunction results in defects in cell cycle progression in mNPCs.** (**A**) Time series of mNPCs in vitro using Viafluor-647 live cell microtubule staining kit (Biotium, #70063) in shCtrl and shArl2. (**B**) Quantification graph showing the average time taken for a single mitotic cycle in control = 1.32 ± 0.21 h and shArl2 = 6.55 ± 1.52 h. (**C**) Time series of mNPCs in vitro using Viafluor-488 live cell microtubule staining kit (Biotium, #70062) in Arl2$^{WT}$, Arl2$^{Q70L}$, and Arl2$^{T30N}$. (**D**) Quantification graph showing the average time taken for a single mitotic cycle in control = 1.10 ± 0.39 h, Arl2$^{WT}$ = 1.09 ± 0.51 h, Arl2$^{Q70L}$ = 1.17 ± 0.41 h, Arl2$^{T30N}$ = 5.48 ± 2.64 h in mNPCs overexpressing Arl2$^{WT}$, Arl2$^{Q70L}$, and Arl2$^{T30N}$. Scale bars; A and C = 10 μm. Source data can be found in S1 Data. Arl2, ADP ribosylation factor-like GTPase 2; mNPC, mouse neural progenitor cell.

caused a significant increase in the mitotic duration as compared to the control (Fig 4C and 4D and S4 Movie; control = 1.10 ± 0.39, Arl2$^{WT}$ = 1.09 ± 0.39, Arl2$^{T30N}$ = 5.48 ± 2.63, Arl2$^{Q70L}$ = 1.17 ± 0.40 h), suggesting an additional defect in the cell cycle progression caused by this mutant form. Consistent with this result, upon Arl2 KD at E14, 1 day after IUE, there was a significant increase in the proportion of Caspase-3$^{+}$ GFP$^{+}$ cells in the VZ and SVZ (15.7 ± 1.85%) as compared to control (2.65 ± 0.80%) (S2E and S2F Fig), suggesting that defects in cell cycle progression may lead to an increase in cell death of mNPCs in vivo.

## Loss of Arl2 results in a significant reduction of centrosomal microtubule growth in mNPCs in vitro

Arl2 regulates centrosomal microtubule nucleation and growth in various cell types including *Drosophila* NSCs [21,22,24]. We sought to investigate whether mouse Arl2 plays a conserved role in microtubule growth in mNPCs. To this end, we performed microtubule regrowth assay, wherein mNPCs were synchronized in the S phase using thymidine and microtubules were depolymerized by nocodazole treatment in these cells. Microtubule regrowth labelled by α-tubulin was assessed in a time-course experiment following washout of nocodazole (Fig 5A). Before nocodazole treatment, shArl2 cells showed a slight but not significant reduction of microtubule intensity (89.59 A.U.) as compared to control (Fig 5B and 5C, 116.95 A.U.). Microtubules were efficiently depolymerized in both control and Arl2 KD cells (t = 0), as only weak residual microtubules labelled by α-tubulin were seen at the centrosome following noco- dazole treatment (Fig 5B and 5C, shCtrl: 43.15 A.U. versus shArl2: 29.21 A.U.). In control, robust microtubules were observed around the centrosome, at various time points following recovery (Fig 5B and 5C). In contrast, Arl2 KD in mNPCs reassembled less microtubule mass after 5-min recovery (Fig 5B and 5C; shCtrl: 72.06 A.U. versus shArl2: 38.16 A.U.) and after 10 min (Fig 5B and 5C; shCtrl: 89.61 A.U. versus shArl2: 51.09 A.U.). Interestingly, even after 30 min of recovery, Arl2 KD cells were still unable to recover their microtubule mass as compared to control (Fig 5B and 5C; shCtrl: 103.13 A.U. versus shArl2: 53.83 A.U.). These results suggest that Arl2 promotes microtubule growth in mNPCs.

To further determine the role of Arl2 in microtubule growth of mNPCs, we performed live imaging to track the growing ends of microtubules by using the plus-end microtubule binding protein EB3, which is enriched in the nervous system [32]. Microtubules have a fast growing plus end and a slow growing minus end, which contribute to its dynamics. EB3-GFP has been widely used to track microtubule dynamics, as it marks the microtubule growth and dynamics by binding to the plus-ends of microtubules. EB3-GFP velocity and density monitored in live imaging are good representations for microtubule dynamics [33]. Remarkably, KD Arl2 (shArl2 marked by GFP) in mNPCs expressing EB3-tdTomato resulted in a significant reduc- tion in velocity of both anterograde (0.076 μm/s) and retrograde EB3 comets (0.048 μm/s) as compared to control (Fig 5D–5G and S5 Movie; anterograde, 0.092 μm/s; retrograde, 0.066 μm/s). Furthermore, the total density of EB3 comets was notably reduced as compared to the control (Fig 5D, 5E and 5H and S5 Movie; shCtrl = 0.30 No./μm$^2$; shArl2 = 0.24 No./ μm$^2$). Taken together, our data suggest that Arl2 is critical for microtubule growth in mNPCs.

## Mitochondria defect is not the major cause of neurogenesis deficit observed in Arl2 dysfunction

Given the importance of Arl2 in regulating both microtubule growth and mitochondrial func- tions [21,30], we sought to pinpoint the mechanism underlying the role of Arl2 during neuro- genesis. To this end, we generated the Lys71 to Arg71 (Arl2$^{K71R}$) mutation in mouse Arl2, which is known to cause mitochondrial fragmentation and immobility without disruption in

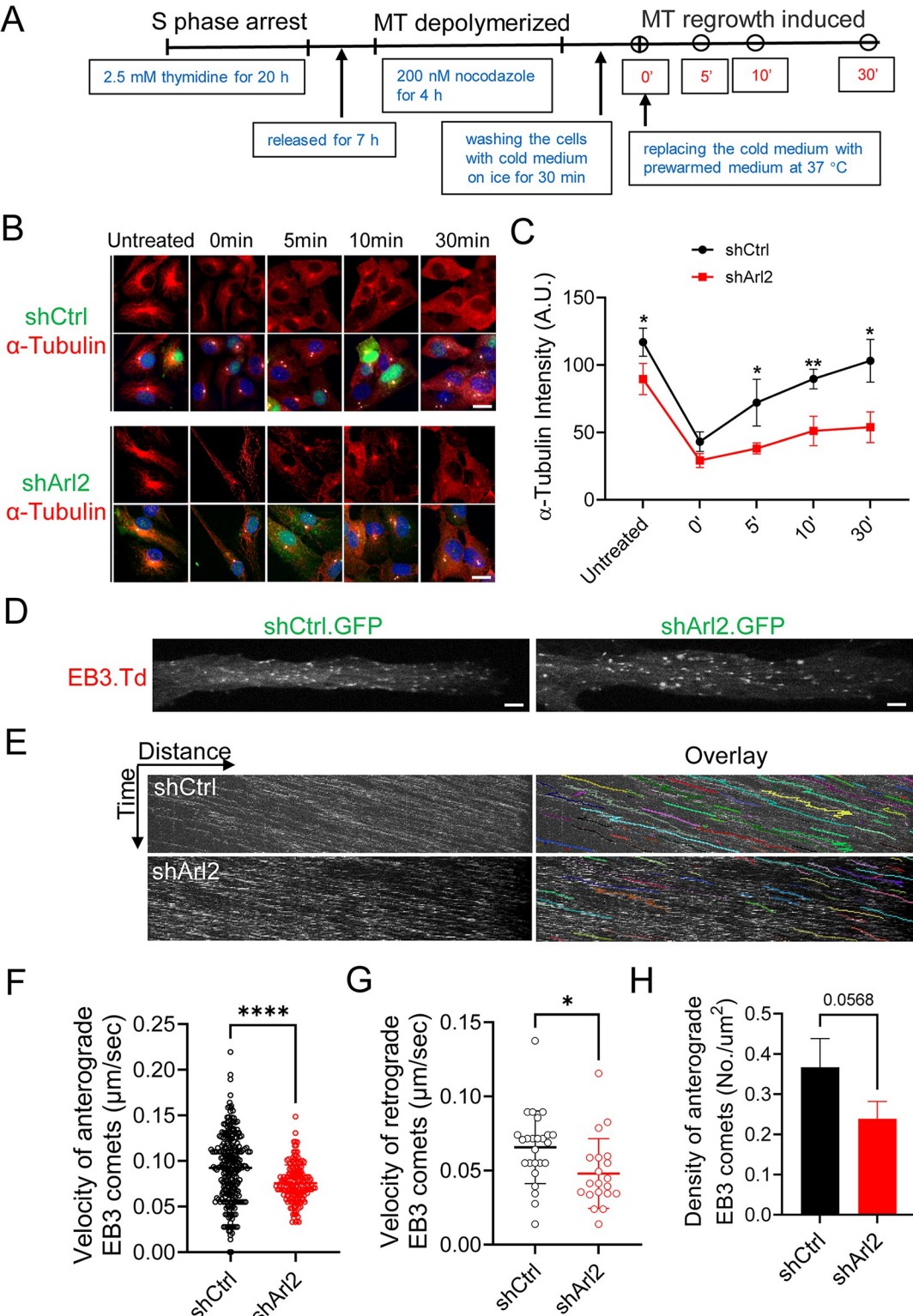

**Fig 5. Loss of Arl2 results in a significant reduction of centrosomal microtubule growth in mNPCs in vitro.** (**A**) Schematic representation of centrosomal microtubule regrowth assay. (**B**) Immunostaining micrographs showing the microtubule regrowth labelled by α-tubulin within the time course (0, 5, 10, and 30 min) in both shCtrl and shArl2 groups. (**C**) Line graph representing α-tubulin intensity in shArl2 group (untreated = 89.59 ± 11.55, 0 min = 29.21 ± 5.16, 5 min = 38.16 ± 4.07, 10 min = 51.09 ± 10.80, and 30 min = 53.83 ± 11.31) compared to the control (untreated = 116.95 ± 10.39, 0 min = 43.14 ± 7.23, 5

min = 72.06 ± 17.30, 10 min = 89.60 ± 7.26, and 30 min = 103.13 ± 15.88) (unit = A.U.). The values represent the mean ± SD. Multiple $t$ test in C, $n$ = 3. Differences were considered significant at *$p < 0.05$, **$p < 0.01$. (**D**) Live imaging micrograph to track the growing ends of microtubules by using the plus-end microtubule binding protein EB3 tagged with Tdtomato (Td) in mNPCs in both shCtrl and shArl2 groups. (**E**) Kymographs showing the EB3-Td comets movement in mNPCs in both shCtrl and shArl2 groups. (**F, G**, and **H**) Quantification graphs representing the velocity of anterograde EB3 comets (shCtrl: 0.092 ± 0.036 μm/s vs. shArl2: 0.076 ± 0.020 μm/s), the velocity of retrograde EB3 comets (shCtrl: 0.066 ± 0.024 μm/s vs. shArl2: 0.048 ± 0.023 μm/s) and the total density of EB3 comets (shCtrl: 0.30 ± 0.046 No./μm² vs. shArl2: 0.24 ± 0.043 No./μm²). The values represent the mean ± SD. Student $t$ test in F, G, and H, $n$ = 3. Differences were considered significant at *$p < 0.05$, ***$p < 0.001$. Scale bars; B = 10 μm, D = 1 μm. Source data can be found in S1 Data. Arl2, ADP ribosylation factor-like GTPase 2; mNPC, mouse neural progenitor cell.

microtubule assembly [34]. As expected, Arl2$^{K71R}$ overexpression showed fragmented mitochondria with shortened mitochondrial length (2.41 ± 0.99 μm) as compared to control (5.53 ± 3.78 μm; S5D and S5E Fig) in mouse NPCs in vitro. Interestingly, at day 3 after IUE during in vivo cortical development, overexpression of mouse Arl2$^{K71R}$ mutant resulted in the migration of Td+ cells to the CP (VZ = 12.53 ± 3.09%, SVZ = 17.44 ± 2.59%, IZ = 37.31 ± 2.35%, CP = 32.96 ± 5.10%), undistinguishable from mouse Arl2$^{WT}$ overexpression (VZ = 14.79 ± 1.84%, SVZ = 18.57 ± 2.28%, IZ = 35.32 ± 1.04%, CP = 31.30 ± 2.14%; S5F and S5G Fig), which mimicked the effect of overexpression of the wild-type Arl2 on cortical development (Fig 3).

Consistent with previous reported role of Arl2 in mitochondria fusion [26], Arl2$^{Q70L}$ overexpression in mNPCs caused a dramatic decrease in the number of cells with tubular mitochondria, indicating an increase in mitochondria fusion, while Arl2$^{T30N}$ overexpression resulted in a severe mitochondrial fragmentation as compared to the control or Arl2$^{WT}$ overexpression (S6A and S6B Fig and S6 Movie). Furthermore, neither shArl2 KD nor overexpression of Arl2 showed any change in mitochondrial morphology in mouse brains 4 days after IUE as compared to control (S6C Fig). Taken together, mitochondrial defects were not the primary cause for migration deficits observed in Arl2 dysfunction.

## Overexpression of Arl2 mutant forms leads to defects in microtubule growth in mNPCs in vitro

Next, we performed microtubule regrowth assay for mNPCs overexpressing Arl2$^{WT}$, Arl2$^{Q70L}$, and Arl2$^{T30N}$. Both control and Arl2$^{WT}$ cells treated with nocodazole (t = 0 s) showed weak residual microtubules at the centrosome, suggesting an efficient microtubule depolymerization (Fig 6A and 6B, control: 49.71 ± 7.36%; Arl2$^{WT}$: 45.57 ± 6.61%). Remarkably, after 5 min of recovery, Arl2$^{WT}$ mNPCs displayed more abundant microtubule density than control (Fig 6A and 6B, control: 59.58 ± 14.29% A.U.; Arl2$^{WT}$: 73.33 ± 17.61% A.U.). Even after 30 min of recovery, Arl2$^{WT}$ mNPCs still showed significantly higher microtubule density as compared to control (Fig 6A and 6B, control: 96.95 ± 16.54% A.U.; Arl2$^{WT}$: 113.92 ± 11.01% A.U.), suggesting that overexpression of Arl2$^{WT}$ likely leads to overgrowth of microtubules in mNPCs. Similarly, overexpression of Arl2$^{K71R}$ caused a significant increase in overall microtubule density (0 min = 64.63 ± 16.30 A.U., 5 min = 87.12 ± 11.05 A.U., 10 min = 118.77 ± 22.48 A.U., and 30 min = 109.60 ± 20.81 A.U.), as compared to control recovery (0 min = 33.72 ± 6.30 A.U., 5 min = 66.10 ± 14.74 A.U., 10 min = 75.92 ± 11.53 A.U., and 30 min = 86.86 ± 15.31 A.U.; Fig 6C), suggesting that Arl2$^{K71R}$ behaves as the Arl2 wild-type form. In contrast, overexpression of Arl2$^{Q70L}$ and Arl2$^{T30N}$ in mNPCs reassembled significantly lesser microtubule mass, at various time points following recovery as compared to control (Fig 6A and 6B; Arl2$^{T30N}$, untreated = 74.08 ± 1.98 A.U., 0 min = 39.85 ± 2.09 A.U., 5 min = 43.91 ± 1.82 A.U., 10 min = 30.11 ± 5.37 A.U., and 30 min = 38.74 ± 12.51 A.U.; Arl2$^{Q70L}$, untreated = 79.08 ± 11.56 A.U., 0 min = 28.50 ± 4.00 A.U., 5 min = 49.51 ± 23.50 A.U., 10 min = 42.23 ± 7.43 A.U., and

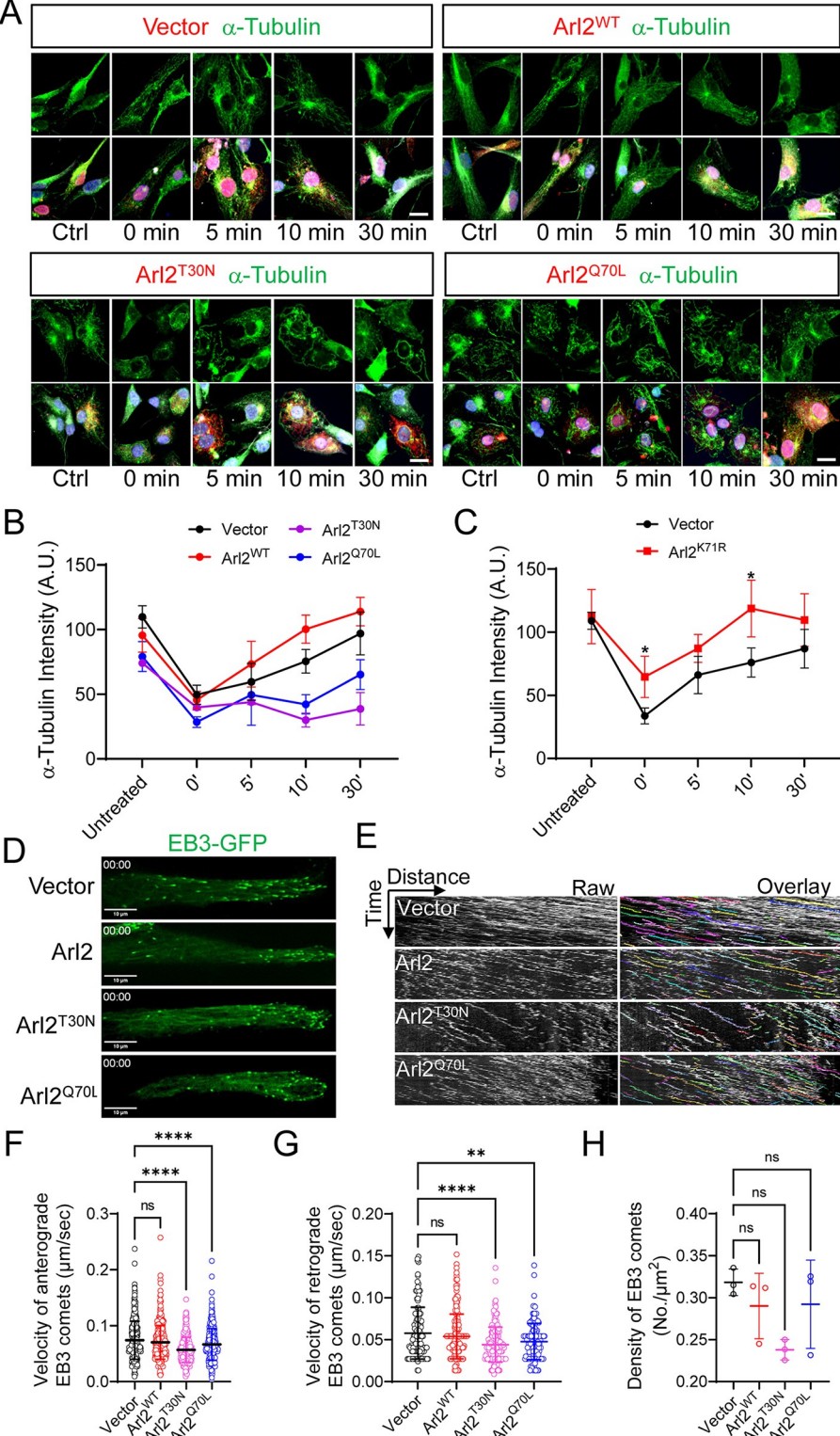

**Fig 6. Overexpression of mutant forms of Arl2 leads to defects in microtubule growth in mNPCs. (A)** Immunostaining micrographs showing microtubule regrowth labelled by α-tubulin within the time course (0, 5, 10, and 30 min) in Arl2^WT, Arl2^T30N, and Arl2^Q70L groups in mNPCs. (**B**) Line graph representing α-tubulin intensity in Arl2^WT group (untreated = 95.57 ± 13.03, 0 min = 45.57 ± 6.61, 5 min = 73.33 ± 17.61, 10 min = 100.32 ± 10.82, and 30 min = 113.92 ± 11.01), Arl2^T30N group (untreated = 74.08 ± 1.98, 0 min = 39.85 ± 2.09, 5 min = 43.91 ± 1.82, 10

min = 30.11 ± 5.37, and 30 min = 38.74 ± 12.51), Arl2$^{Q70L}$ group (untreated = 79.08 ± 11.56, 0 min = 28.50 ± 4.00, 5 min = 49.51 ± 23.50, 10 min = 42.23 ± 7.43, and 30 min = 65.19 ± 11.47) compared to the control (untreated = 109.78 ± 8.61, 0 min = 49.71 ± 7.36, 5 min = 59.58 ± 14.29, 10 min = 75.43 ± 9.13, and 30 min = 96.95 ± 16.54) in mNPCs. The values represent the mean ± SD, unit = A.U. Two-way ANOVA with multiple comparisons in B, $n = 3$. Differences were considered significant at $^*p < 0.05$, $^{**}p < 0.01$, $^{***}p < 0.001$, and $^{****}p < 0.0001$, ns = nonsignificance. (C) Line graph of microtubule regrowth assay representing α-tubulin intensity in overexpression of Arl2$^{K71R}$ mutant group (untreated = 112.39 ± 21.47, 0 min = 64.63 ± 16.30, 5 min = 87.12 ± 11.05, 10 min = 118.77 ± 22.48, and 30 min = 109.60 ± 20.81) as compared to the control (untreated = 108.99 ± 6.73, 0 min = 33.72 ± 6.30, 5 min = 66.10 ± 14.74, 10 min = 75.92 ± 11.53, and 30 min = 86.86 ± 15.31) (unit = A.U.) in mNPCs in vitro. Multiple $t$ test in C, $n = 3$. Differences were considered significant at $^*p < 0.05$, ns = nonsignificance. (D) Live imaging micrograph to track the growing ends of microtubules by using the plus-end microtubule binding protein EB3 tagged with GFP in mNPCs in Arl2$^{WT}$, Arl2$^{T30N}$, and Arl2$^{Q70L}$ groups. (E) Kymographs showing the EB3-GFP comets movement in mNPCs in Arl2$^{WT}$, Arl2$^{T30N}$, and Arl2$^{Q70L}$ groups. (F, G, and H) Quantification graphs representing the velocity of anterograde EB3 comets (control: 0.074 ± 0.034 μm/s, Arl2$^{WT}$: 0.070 ± 0.03 μm/s; Arl2$^{T30N}$: 0.057 ± 0.022 μm/s and Arl2$^{Q70L}$: 0.067 ± 0.028 μm/s), the velocity of retrograde EB3 comets (control: 0.058 ± 0.031 μm/s, Arl2$^{WT}$: 0.054 ± 0.027 μm/s; Arl2$^{T30N}$: 0.044 ± 0.021 μm/s and Arl2$^{Q70L}$: 0.048 ± 0.021 μm/s) and the total density of EB3 comets (control: 0.32 ± 0.02 No./μm$^2$, Arl2$^{WT}$: 0.29 ± 0.04 No./μm$^2$; Arl2$^{T30N}$: 0.24 ± 0.01 No./μm$^2$ and Arl2$^{Q70L}$: 0.29 ± 0.05 No./μm$^2$, $n = 3$). The values represent the mean ± SD. One-way ANOVA in E, F, and G. Differences were considered significant at $^{**}p < 0.01$ and $^{****}p < 0.0001$, ns = nonsignificance. Scale bars; A = 5 μm; D = 10 μm. Source data can be found in S1 Data. Arl2, ADP ribosylation factor-like GTPase 2; GFP, green fluorescent protein; mNPC, mouse neural progenitor cell.

30 min = 65.19 ± 11.47 A.U.). These results further support that Arl2 promotes microtubule growth in mNPCs.

To further analyze the effect of Arl2 on microtubule growth, we performed live imaging in mNPCs overexpressing Arl2$^{WT}$, Arl2$^{Q70L}$, or Arl2$^{T30N}$. Remarkably, overexpression of Arl2$^{WT}$ showed a significant increase in the intensity of microtubules as compared to control (S6A Fig). In contrast, both Arl2$^{Q70L}$ and Arl2$^{T30N}$ showed a significant reduction in the intensity of microtubules as compared to control (S6A Fig).

To further validate the role of Arl2 in microtubule growth, we performed live imaging of EB3-GFP comets in Arl2$^{WT}$, Arl2$^{Q70L}$, and Arl2$^{T30N}$ mNPCs. Similar to Arl2 KD, overexpression of Arl2$^{Q70L}$ and Arl2$^{T30N}$ resulted in a significant reduction in velocity in anterograde (Arl2$^{T30N}$, 0.057 ± 0.022 μm/s; Arl2$^{Q70L}$, 0.067 ± 0.028 μm/s) and retrograde EB3 comets (Arl2$^{T30N}$, 0.044 ± 0.021 μm/s and Arl2$^{Q70L}$, 0.048 ± 0.021 μm/s) as compared to control (Fig 6D–6G and S7 Movie; anterograde, (0.074 ± 0.034 μm/s), retrograde, (0.058 ± 0.031 μm/s). The total density of EB3 comets appeared to be normal by overexpression of Arl2$^{WT}$, Arl2$^{Q70L}$, or Arl2$^{T30N}$ (Fig 6D, 6E and 6G and S7 Movie). Furthermore, overexpression of Arl2$^{WT}$ showed no obvious change in movements of EB3-GFP comets (Fig 6E and 6H), likely due to its weaker effects than overexpression Arl2 mutant forms. Taken together, these observations indicate that Arl2 regulates microtubule growth in mNPCs.

## Arl2 localizes to the PCM of the centrosomes and facilitates γ-tubulin localization at the centrosomes in mNPCs

Arl2 localizes to the centrosomes in different cell types including HEK and CHO cells and presumably localizes to PCM [20]. To determine whether Arl2 localizes to the PCM in HEK293 cells, we examined the ultrastructure of Arl2 (Arl2-HA) and Cdk5rap2 (Cdk5rap2-Myc), a centrosomal protein that is involved in microtubule organization [35], using super-resolution microscopy (Fig 7A). Remarkably, Arl2 and Cdk5rap2 formed ring-like structures that colocalized with one another at the centrosome colabelled by γ-tubulin in interphase and metaphase cells (Fig 7A). This observation suggests that Arl2 indeed localizes to the PCM. Furthermore, knocking down of Arl2 resulted in a significant decrease in γ-tubulin intensity at the centrosomes in metaphase of mNPCs (Fig 7B and 7D; 68.59 ± 9.31 A.U.) and interphase of mNPCs (S7C and S7E Fig; 52.83 ± 10.22 A.U.) as compared to control (metaphase: 114.9 ± 24.88 A.U.,

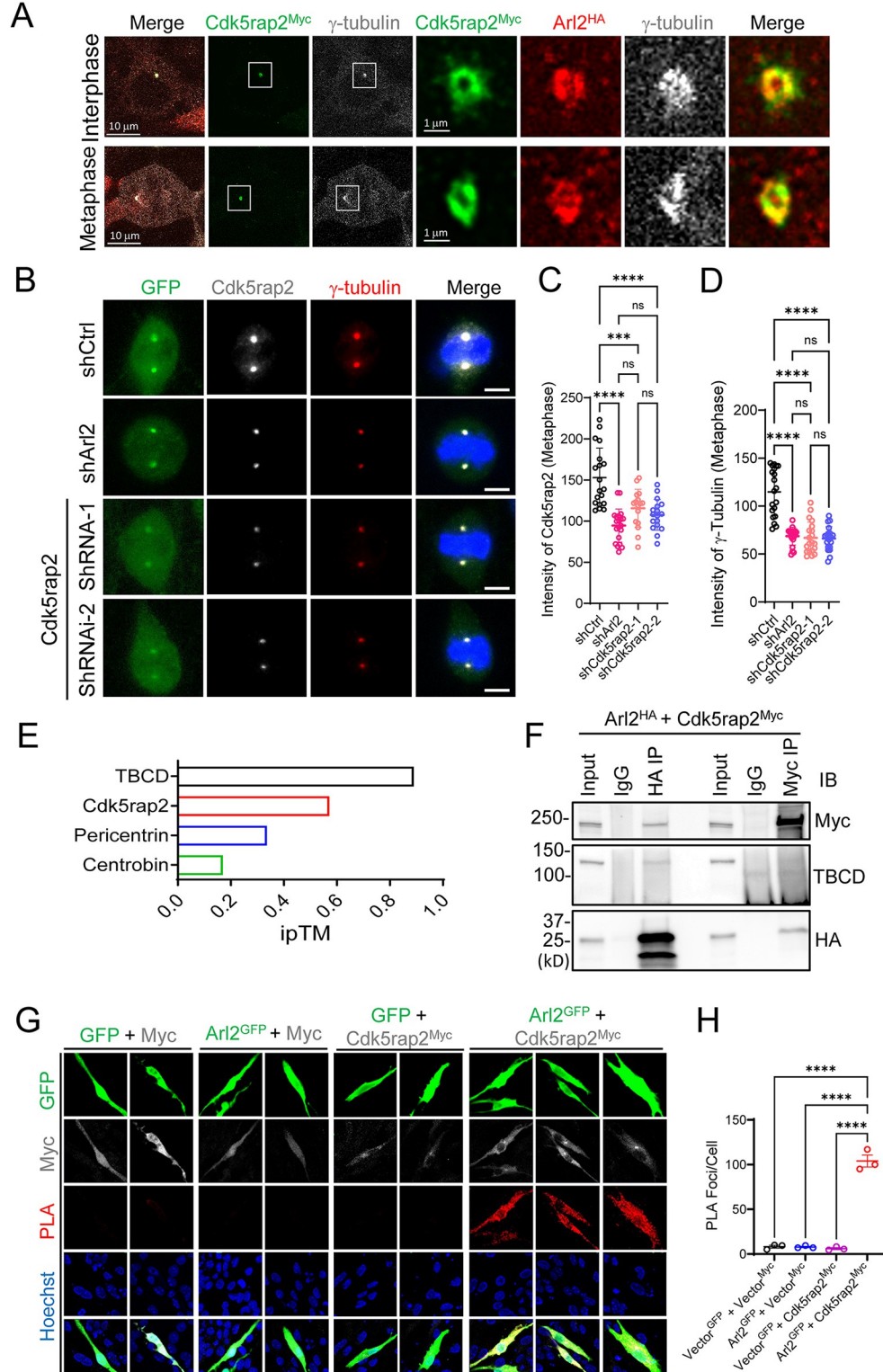

**Fig 7. Arl2 localizes to the centrosomes and is required for γ-tubulin localization at the centrosomes in mNPCs.**
(**A**) Immunostaining micrographs of HEK293 cells cotransfected with Arl2-HA and Cdk5rap2-Myc were imaged using super-resolution microscopy labelled for γ-tubulin, Myc, and HA. (**B**) Immunostaining micrographs showing Cdk5rap2 and γ-tubulin in shCtrl, shArl2, and shCdk5rap2 groups in mNPCs. (**C**) Quantification graph showing Cdk5rap2 intensity at metaphase of mNPCs (94.5 ± 20.06, *n* = 3 batches with 20 cells) upon Arl2 KD and

(shCdk5rap2-1 = 115.6 ± 23.13, $n$ = 3 batches with 16 cells; shCdk5rap2-2 = 107.5 ± 18.77, $n$ = 3 batches with 17 cells) upon Cdk5rap2 KD as compared to shCtrl (153.0 ± 35.96, $n$ = 3 batches with 19 cells). (**D**) Quantification graph showing γ-tubulin intensity at metaphase of mNPCs (68.59 ± 9.31, $n$ = 3 batches with 20 cells) upon Arl2 KD and (shCdk5rap2-1 = 66.85 ± 16.44, $n$ = 3 batches with 20 cells; shCdk5rap2-2 = 66.11 ± 12.33, $n$ = 3 batches with 20 cells) upon Cdk5rap2 KD as compared to shCtrl (114.9 ± 24.88, $n$ = 3 batches with 20 cells). The values represent the mean ± SD. One-way ANOVA in C and D. Differences were considered significant at ***$p < 0.001$ and ****$p < 0.0001$, ns = nonsignificance. (**E**) Bar graph showing AlphaFold multimer interaction prediction of Arl2 and Cdk5rap2, Arl2 and Pericentrin (PCNT), Arl2 and Centrobin with an ipTM score of 0.57, 0.34, 0.17, respectively, compared with TBCD, a known interactor of Arl2 with an ipTM score of 0.89. (**F**) Co-immunoprecipitation by overexpressing Arl2 (HA-Arl2) and Cdk5rap2 (Myc-Cdk5rap2) in HEK293 cells. Following precipitation with a HA antibody, the resulting protein complexes exhibited an anticipated 37 kD band corresponding to HA-Arl2 as well as 250 kD band corresponding to Myc-Cdk5rap2. TBCD was used as positive control, which also co-immunoprecipitated following precipitation with a HA antibody. Similarly, following precipitation with Myc antibody, bands corresponding Myc-Cdk5rap2 and HA-Arl2 were observed. (**G**) PLA showing overexpressing Arl2 (Arl2-GFP) and Cdk5rap2 (Myc-Cdk5rap2) (Vector-GFP and Myc-Vector, Arl2-GFP and Myc-Vector, Vector-GFP and Myc-Cdk5rap2, Arl2-GFP and Myc-Cdk5rap2) in mNPCs. (**H**) Quantification graph of the PLA foci per cell with no red dot, weak red dots, and strong red dots for (G). Vector-GFP and Myc-Vector, 8.17 ± 2.75; Arl2-GFP and Myc-Vector, 7.83 ± 1.44; Vector-GFP and Myc-Cdk5rap2, 6.11 ± 1.99; Arl2-GFP and Myc-Cdk5rap2, 104.00 ± 11.53; $n$ = 3). The values represent the mean ± SD. One-way ANOVA in H. Differences were considered significant at ****$p < 0.0001$. Scale bars; A = 10 μm; boxed image for A = 1 μm; B = 5 μm; G = 40 μm. Source data can be found in S1 Data. Arl2, ADP ribosylation factor-like GTPase 2; ipTM, interface pTM; KD, knockdown; mNPC, mouse neural progenitor cell; PLA, proximity ligation assay; TBCD, Tubulin folding cofactor D.

interphase: 80.47 ± 23.81 A.U., respectively). These data suggest that Arl2 is a centrosomal protein required for centrosomal assembly in mNPCs.

## Arl2 interacts with the centrosomal protein Cdk5rap2

Next, we explored whether Arl2 and Cdk5rap2 can interact with each other. AlphaFold multimer is emerging as a powerful and accurate approach for in silico prediction of protein–protein interactions based on deep learning method [36–38]. Using AlphaFold multimer, Tubulin folding cofactor D (TBCD), a known strong interactor of Arl2, had an interface pTM (ipTM) score of 0.89, indicating the reliability of this approach. We found that Cdk5rap2 had an ipTM score of 0.57, suggesting that Cdk5rap2 is a strong candidate of Arl2-interacting protein (Fig 7E). Pericentrin, another centrosomal protein required for cortical development, also interacts with and recruits Cdk5rap2 to the centrosome in the mNPCs [16]. Interestingly, AlphaFold multimer also predicts that Arl2 can potentially interact with Pericentrin with an ipTM score of 0.33 (Fig 7E). Centrobin, a centriolar protein as a negative control for the interaction testing, was predicted not to interact with Cdk5rap2 with an ipTM score of 0.17 by AlphaFold multimer (Fig 7E).

To validate the predicted interaction between Arl2 and Cdk5rap2, we performed co-immunoprecipitation by overexpressing Arl2 (HA-Arl2) and Cdk5rap2 (Myc-Cdk5rap2) in HEK293 cells (Fig 7F). Following immunoprecipitation with a HA antibody, the resulting protein complexes exhibited an anticipated 37 kD band corresponding to HA-Arl2 as well as 250 kD band corresponding to Myc-Cdk5rap2, suggesting that Arl2 and Cdk5rap2 physically associate with each other (Fig 7F). TBCD was used as positive control, which also co-immunoprecipitated following precipitation with a HA antibody (Fig 7F). Similarly, following precipitation with Myc antibody, bands corresponding Myc-Cdk5rap2 and HA-Arl2 were observed further confirming the interaction between Cdk5rap2 and Arl2 (Fig 7F).

To further validate the association between Arl2 and Cdk5rap2, we employed proximity ligation assay (PLA), a technique that enables the detection of protein–protein interactions with high specificity and sensitivity [39]. We coexpressed various proteins tagged with Myc or GFP in mNPCs and quantified PLA foci that indicated protein–protein interactions (Fig 6G and 6H). The vast majority of mNPCs coexpressing both Myc and GFP controls displayed

weak fluorescence signal of merely a few PLA foci (Fig 7G and 7H; 8.17 ± 2.75). Similarly, the vast majority of cells coexpressing Arl2-GFP with control Myc or Myc-Cdk5rap2 with control GFP displayed few PLA puncta per cell 7.83 ± 1.44; Vector-GFP and Myc-Cdk5rap2, 6.11 ± 1.99), under each coexpression condition, respectively (Fig 7G and 7H). By contrast, mNPCs coexpressing Arl2-GFP and Myc-Cdk5rap2 displayed strong signal with a plethora of PLA foci (Fig 7G and 7H; 104 ± 11.5).

Taken together, our data indicate that Arl2 and Cdk5rap2 can physically interact with each other.

## Cdk5rap2 affects neuronal migration and proliferation in vitro and in vivo, similar to Arl2 loss-of-function

Cdk5rap2 maintains NPC pool in the developing Neocortex [16]. We found similar neurogenesis defects following knocking down of Cdk5rap2 (S7 Fig). Upon Cdk5rap2 KD in mNPCs by 2 independent shCdk5rap2-1 and shCdk5rap2-2 tagged with GFP, Cdk5rap2 protein level detected by anti-Cdk5rap2 antibodies in WB was reduced to 26% in shCdk5rap2-1 and 32% in shCdk5rap2-2, respectively, compared with the control (S7A and S7B Fig), suggesting efficient KDs by both shCdk5rap2. Silencing endogenous Cdk5rap2 expression in the primary culture of mNPCs in vitro by lentivirus (pPurGreen) infection in 48-h culture resulted in a significant decrease in the intensity of Cdk5rap2 at the centrosomes in metaphase mNPCs (shCdk5rap2-1 = 115.6 A.U.; shCdk5rap2-2 = 107.5 A.U., respectively) as compared to control (Fig 7B and 7C; metaphase: 153.0 A.U.). Similarly, the intensity of Cdk5rap2 at the centrosomes in interphase mNPCs (shCdk5rap2-1 = 34.73 A.U.; shCdk5rap2-2 = 44.53 A.U., respectively) was significantly reduced as compared to control (S7C and S7D Fig; interphase: 77.37 A.U.). Furthermore, there was a significant reduction in γ-tubulin intensity upon Cdk5rap2 KD (shCdk5rap2-1 = 66.85 A.U.; shCdk5rap2-2 = 66.11 A.U., respectively) in metaphase mNPCs as well as at the centrosomes in interphase mNPCs (shCdk5rap2-1 = 32.02 A.U.; shCdk5rap2-2 = 36.29 A.U., respectively) as compared to control (metaphase: 114.9 A.U.; interphase: 80.47 A.U.; Figs 7B, 7C, S7C and S7D). Remarkably, the proportion of EdU+ cells in the primary culture of mNPCs was dramatically reduced upon Cdk5rap2 KD (shCdk5rap2-1 = 26.42 ± 7.89%; shCdk5rap2-2 = 34.58 ± 5.72%, respectively) as compared to the control group (shCtrl = 58.84 ± 6.06%) (S7F and S7G Fig).

We introduced Cdk5rap2 shRNA-1 via microinjection into the lateral ventricle of mouse embryos, followed by IUE at E13. At E14, 1 day after IUE and following 6-h pulse-labelling with EdU before sample collection, we observed a substantial reduction in the proportion of EdU+/GFP+ double-labelled cells in the shCdk5rap2 group 47.05 ± 5.43% as compared to the control group 58.92 ± 3.62% (Fig 8A and 8B). Remarkably, at E17, 4 days after IUE, Cdk5rap2 KD resulted in a significant number of GFP+ cells to persist in the VZ + SVZ (13.79 ± 2.85%) and IZ (27.40 ± 2.62%) with fewer GFP+ cells migrating towards the CP (58.80 ± 2.17%) as compared to control (S7H and S7I Fig, control, VZ + SVZ (4.20 ± 2.73%), IZ (14.29 ± 2.87%), CP (81.51 ± 3.26%). These data suggest that loss of Cdk5rap2 affects neuronal migration and proliferation in vitro and in vivo, similar to Arl2 loss-of-function.

## Arl2 is required for the centrosomal localization of Cdk5rap2 in mNPCs

Since we demonstrate that Arl2 colocalizes and physically associates with Cdk5rap2 at the centrosomes of mNPCs (Fig 7), we wondered whether Arl2 is required for the centrosomal localization of Cdk5rap2 in mNPCs. Indeed, Cdk5rap2 centrosomal localization was diminished upon Arl2 KD in mNPCs at interphase (S7C and S7D Fig) and mitosis (Fig 7B and 7C). Furthermore, Cdk5rap2 intensity were significantly reduced upon Arl2 KD in mNPCs at

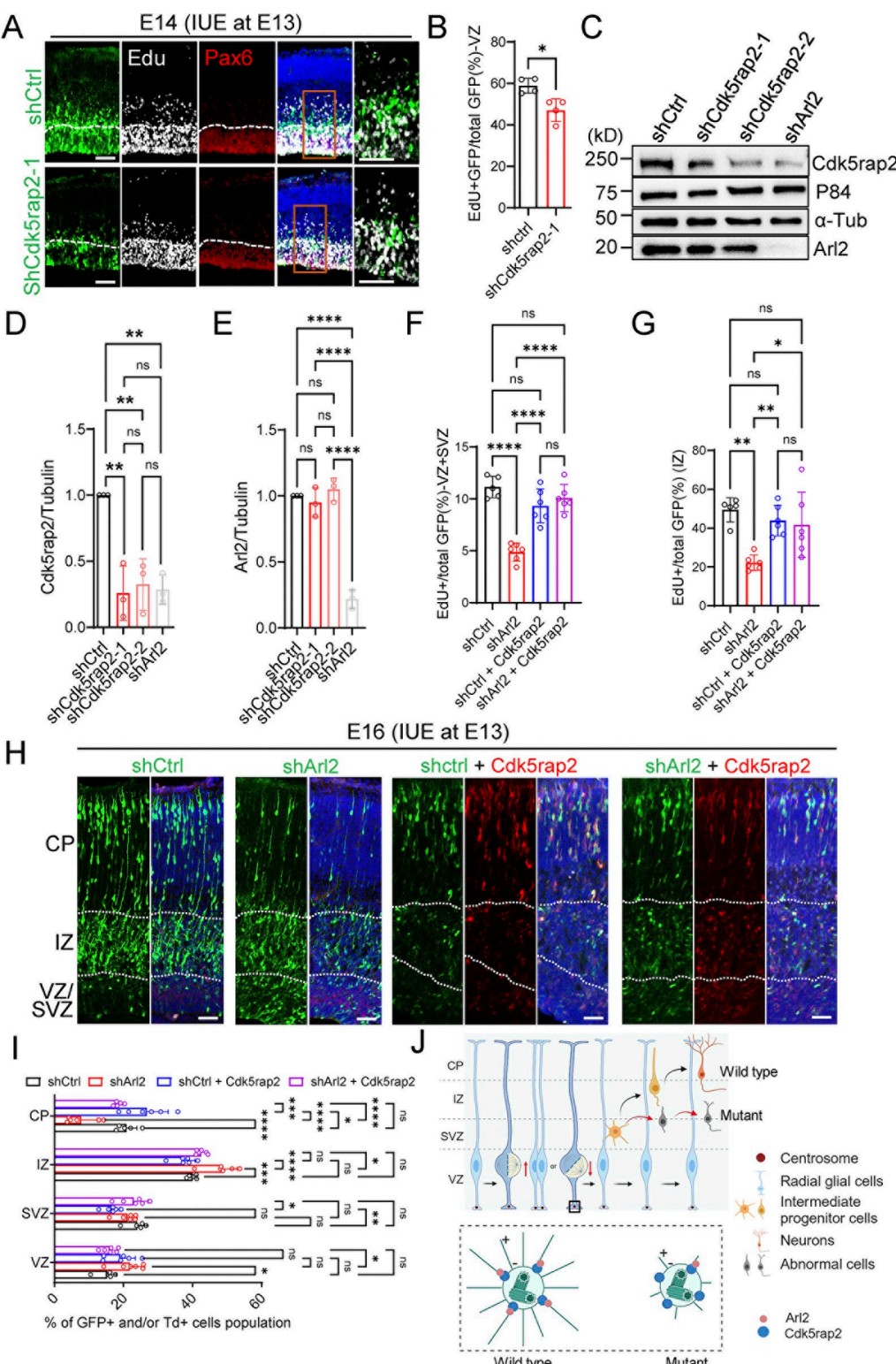

**Fig 8. Arl2 functions upstream of Cdk5rap2 in regulating neuronal cell migration and proliferation in the developing cortex.** (**A**) Brain slices from shCtrl (scrambled control) and shCdk5rap2 (Cdk5rap2 shRNA) groups at E14, 1 day after IUE, were labelled with EdU and GFP. (**B**) Bar graphs showing reduced EdU incorporation upon Cdk5rap2 KD (47.05 ± 5.43% in shCdk5rap2 vs. 58.92 ± 3.62% in shCtrl). The values represent the mean ± SD. Student t test in C, n = 4 embryos. Differences were considered significant at *p < 0.05. (**C**) WB analysis of mNPC protein extracts of

control (H1-shCtrl-GFP), Cdk5rap2 KD (H1-shCdk5rap2-GFP), and Arl2 KD with lentivirus (pPurGreen) infection in 48-h culture. Blots were probed with anti-Cdk5rap2 and anti-Arl2 antibody, α-Tubulin and p84 as loading control. (**D**) Bar graphs representing Cdk5rap2 protein levels upon Arl2 KD (shCdk5rap2-1 = 0.26 ± 0.21 and shCdk5rap2-2 = 0.32 ± 0.20; shArl2 = 0.29 ± 0.11 normalized in shCtrl, $n$ = 3) in mNPCs. (**E**) Bar graphs representing Arl2 protein levels upon Cdk5rap2 KD (shCdk5rap2-1 = 0.95 ± 0.11 and shCdk5rap2-2 = 1.05 ± 0.09; shArl2 = 0.22 ± 0.07 normalized in shCtrl, $n$ = 3) in mNPCs. (**F**) Bar graphs showing the total number of GFP+EdU+ double positive cells in VZ+SVZ by overexpression of Cdk5rap2 in Arl2 KD brains (control = 11.14 ± 1.04%, shArl2 = 4.87 ± 0.81%, shCtrl + Cdk5rap2 = 9.32 ± 1.63%, shArl2 + Cdk5rap2 = 10.08 ± 1.31%) (shCtrl, $n$ = 5 embryos; shArl2, shCtrl + shCdk5rap2, shArl2 + shCdk5rap2, $n$ = 6 embryos) 2 days after IUE. (**G**) Bar graphs showing the total number of GFP+EdU+ double positive cells in IZ by overexpression of Cdk5rap2 in Arl2 KD brains (control = 49.41 ± 6.25%, shArl2 = 22.17 ± 3.98%, shCtrl + Cdk5rap2 = 43.84 ± 7.89%, shArl2 + Cdk5rap2 = 41.68 ± 16.74%) (shCtrl, $n$ = 5 embryos; shArl2, shCtrl + shCdk5rap2, shArl2 + shCdk5rap2, $n$ = 6 embryos) 2 days after IUE. The values represent the mean ± SD. One-way ANOVA in D-G. Differences were considered significant at $^{**}p < 0.01$, $^{****}p < 0.0001$. (**H**) Brain slices from shCtrl, shArl2, shCtrl + Cdk5rap2, and shArl2 + Cdk5rap2 groups at E16, 3 days after IUE, were labelled with GFP (shCtrl, shArl2, shCtrl + Cdk5rap2, and shArl2 + Cdk5rap2) and tdTomato (shCtrl + Cdk5rap2 and shArl2 + Cdk5rap2). (**I**) Bar graphs (images in H) representing GFP+ and/or Td+ cell population in the groups of control (VZ = 15.20 ± 2.75%, SVZ = 23.96 ± 2.88%, IZ = 39.96 ± 1.27%, CP = 20.87 ± 3.01%, $n$ = 5 embryos), shArl2 (VZ = 21.87 ± 4.31%, SVZ = 21.14 ± 2.87%, IZ = 48.45 ± 4.47%, CP = 7.70 ± 4.67%, $n$ = 6 embryos), shCtrl + Cdk5rap2 (VZ = 19.08 ± 4.38%, SVZ = 16.59 ± 2.18%, IZ = 37.63 ± 3.05%, CP = 26.70 ± 6.21%, $n$ = 6 embryos), and shArl2 + Cdk5rap2 (VZ = 16.07 ± 2.15%, SVZ = 22.98 ± 3.99%, IZ = 42.50 ± 1.45%, CP = 18.45 ± 1.15%, $n$ = 6 embryos). (**J**) Working model (made by BioRender): Arl2 plays a novel role in regulating the proliferation and differentiation of mNSCs. Arl2 is required for the proliferation, migration, and differentiation of mouse forebrain NPCs in vitro and in vivo by regulating centrosome assembly and microtubule growth in NPCs. Arl2 physically associates and recruits Cdk5rap2 to the centrosomes to promote microtubule assembly in NPCs. Arl2 functions upstream of Cdk5rap2 in regulating NPC proliferation and migration during mouse cortical development. The values represent the mean ± SD. Two-way ANOVA with multiple comparisons in I. Differences were considered significant at $^{*}p < 0.05$, $^{**}p < 0.01$, $^{***}p < 0.001$, and $^{****}p < 0.0001$, ns = nonsignificance. Scale bar; A and H = 150 μm. Source data can be found in S1 Data. Arl2, ADP ribosylation factor-like GTPase 2; Cdk5rap2, CDK5 Regulatory Subunit Associated Protein 2; CP, cortical plate; EdU, 5-ethynyl-2′-deoxyuridine; GFP, green fluorescent protein; IUE, in utero electroporation; IZ, intermediate zone; KD, knockdown; mNPC, mouse neural progenitor cell; NPC, neural progenitor cell; SVZ, subventricular zone; VZ, ventricular zone; WB, western blot.

interphase (46.14 A.U.) (S7C and S7D Fig) and at metaphase (94.5 A.U.) (Fig 7B and 7C) as compared to the control (interphase: 77.37 A.U.; metaphase: 153.0 A.U.). Moreover, Cdk5rap2 protein levels were significantly reduced upon Arl2 KD in mNPCs by WB analysis (Fig 8C and 8D; shCdk5rap2-1 = 0.26 ± 0.21 and shCdk5rap2-2 = 0.32 ± 0.20; shArl2 = 0.29 ± 0.11 normalized in shCtrl, $n$ = 3). Conversely, Arl2 protein levels in mNPCs were not obviously affected by Cdk5rap2 KD as compared to the control (Fig 8C and 8E, shCdk5rap2-1 = 0.95 ± 0.11 and shCdk5rap2-2 = 1.05 ± 0.09; shArl2 = 0.22 ± 0.07 normalized in shCtrl, $n$ = 3). Thus, Arl2 is required for localization and stabilization of Cdk5rap2 at the centrosomes in mNPCs.

## Cdk5rap2 overexpression rescues neurogenesis defects caused by Arl2 depletion in mouse developing cortex

To determine whether Cdk5rap2 is a physiological relevant target of Arl2 during neurogenesis in vivo, we overexpressed Cdk5rap2 along with Arl2 KD via microinjection into the lateral ventricle of mouse embryos, followed by IUE at E13, and examined cortical neurogenesis. Remarkably, Cdk5rap2 overexpression notably rescued neurogenesis defects caused by Arl2 KD in vivo (Fig 8). Two days after IUE at E15, the total number of GFP+EdU+ double positive cells in VZ+SZ as well as IZ was significantly rescued by overexpression of Cdk5rap2 in Arl2 KD brains (Fig 8F and 8G, VZ+SVZ; control = 11.14 ± 1.04%, shArl2 = 4.87 ± 0.81%, shCtrl + Cdk5rap2 = 9.32 ± 1.63%, shArl2 + Cdk5rap2 = 10.08 ± 1.31%; IZ; control = 49.41 ± 6.25%, shArl2 = 22.17 ± 3.98%, shCtrl + Cdk5rap2 = 43.84 ± 7.89%, shArl2 + Cdk5rap2 = 41.68 ± 16.74%). Furthermore, 3 days after IUE at E16, the number of GFP + cells migrating to the CP in Arl2-depleted mouse brains by overexpressing Cdk5rap2 was dramatically increased to 18.45% compared with 7.70% in Arl2 KD brains (Fig 8H and 8I).

These genetic data further support our model that Arl2 regulates NPC proliferation, migration, and differentiation in mouse cortical development by interacting with Cdk5rap2 to promote microtubule growth from the centrosomes.

## Discussion

In this study, we demonstrate for the first time that the mammalian Arl2, a small GTPase, plays an important role in corticogenesis of the mouse brain. We have identified Arl2 as a new regulator in proliferation and differentiation of mouse NPCs and neuronal migration. Arl2 controls neurogenesis through the regulation of microtubule growth, independent of its function in mitochondrial fusion. We further demonstrate that Arl2 physically associates with Cdk5rap2, a centrosomal protein known to be important for microtubule organization and cortical development. Finally, Arl2 functions upstream of Cdk5rap2 to localize and stabilize Cdk5rap2 at the centrosome to regulate microtubule growth and neuronal migration (Fig 7J). Taken together, our data have identified a novel Arl2-Cdk5rap2 pathway in the regulation of microtubule growth and proliferation of mouse NPCs during cortical development.

### Arl2 regulates mouse corticogenesis via microtubule growth

Although mammalian Arl2 has been shown to be widely expressed in various tissues and is most abundant in the brain [30], the role of mammalian Arl2 in regulating mouse corticogenesis was unknown. In this study, we demonstrate the importance of Arl2 in regulating NPC proliferation and differentiation and neuronal migration during mouse cortical development. We provide evidence that Arl2 is required for centrosome assembly and spindle orientation in NPCs, similar to Cdk5rap2. Our finding is in line with the role of centrosomal/microtubule regulators in spindle orientation of NPCs [40,41]. Our finding also suggests that mouse Arl2 has a novel role in neurite outgrowth in neurons in vitro. All these findings highlight the importance of mouse Arl2 in regulating microtubule growth during corticogenesis.

Similar to the previous finding that transgenic ARL2-Q70L animals exhibit reduced photoreceptor cell function and progressive rod degeneration [42], we found that overexpression of these mutant forms of Arl2 caused cell death of mouse NPCs both in vitro and in vivo. This is consistent with a recent report showing that lengthening mitosis of NPCs resulted in apoptosis of newborn neural progeny [43]. Likewise, human Arl2 plays an essential role for the survival of human embryonic stem cell–derived NPCs [44]. In human brain organoid models, defects in mitosis of NSCs is associated with decrease in stem cell number and apoptosis [18]. Given that our work highlights a novel role of mammalian Arl2 in mouse cortical development in vivo and the conservation of Arl2 in mouse and humans, it will be of great interest to investigate the role of human Arl2 in NPC divisions during cortical development.

It was reported that mitochondria functions are important for radial glia proliferation [45]. Radial glial cells display fused mitochondria, while newborn neurons have highly fragmented mitochondria right after mitosis of NPCs [45]. Increased mitochondria fission promotes neuronal fate, while induction of mitochondria fusion after mitosis redirect daughter cells toward self-renewal [45]. Consistent with previous reports that Arl2 is localized to mitochondria and regulates mitochondria fusion in vitro [26,30,34], we found that $Arl2^{Q70L}$ and $Arl2^{T30N}$ mutant forms have opposing roles in mitochondrial function in mNPCs in vitro. However, neither Arl2 KD nor overexpression of Arl2 obviously altered mitochondrial morphology in the mouse developing brain. In addition, overexpression of $Arl^{K71R}$ mutant, which causes mitochondrial fragmentation without disrupting microtubule assembly [34], behaves similarly to Arl2 wild type in the mNPC proliferation or neuronal migration during neurogenesis in vivo or in our microtubule regrowth assay in vitro. By contrast, Arl2 KD and overexpression of

Arl2$^{Q70L}$ and Arl2$^{T30N}$ mutant forms showed similar phenotypes in proliferation, differentiation, and neuronal migration during neurogenesis in vivo as well as microtubule regrowth assay in vitro. Why overexpression of Arl2$^{Q70L}$ and Arl2$^{T30N}$ mutant forms exhibited the same phenotypes in these processes in mNPCs is unknown. In photoreceptor cells, overexpression of Arl2$^{Q70L}$ binds to ARL2BP and sequester its function, resulting in shortening of cilia in these cells, a phenotype similarly observed in Arl2 KD [42]. It remains to be tested whether the phenotype induced by overexpression of Arl2$^{Q70L}$ in mNPCs is due to sequestering other proteins important for microtubule functions such as ARL2BP. Therefore, the novel role of Arl2 in regulating neurogenesis in the developing cortex is most likely due to its role in microtubule growth, independent of its function in mitochondrial fusion.

## Arl2 plays a novel role in regulating neurogenesis via Cdk5rap2 function

Based on in silico analysis by AlphaFold multimer, co-immunoprecipitation, and PLA, we provide strong evidence that Arl2 physically associates with the centrosomal protein Cdk5rap2. Moreover, our super-resolution imaging clearly shows that Arl2 colocalizes with Cdk5rap2 at the PCM of the centrosomes. Cdk5rap2 is known to regulate centrosomal function and maintain the neural progenitor pool in the developing cortex [16,46]. However, in addition to a similar defect in NPC proliferation upon Cdk5rap2 KD, we observed additional neuronal migration defects following Cdk5rap2 depletion that mimics Arl2 KD. Importantly, Cdk5rap2 overexpression rescues the loss of function phenotype of Arl2 in mice, leading to restored NPC proliferation and neuronal migration to the CP. Therefore, Arl2 functions upstream of Cdk5rap2 in controlling NPC proliferation and neuronal migration via centrosomal functions. Pericentrin, another centrosomal protein required for cortical development, also interacts with and recruits Cdk5rap2 to the centrosome in the mNPCs [16]. Consistent with this finding, our analysis by AlphaFolder multimer also predicts that Arl2 can potentially interact with Pericentrin with an ipTM score of 0.33 (Fig 6E). The current understanding of how Cdk5rap2 protein turnover is regulated, such as through proteasome- or lysosome-mediated degradation, remains unclear. Further investigation is needed to explore the mechanisms governing Cdk5rap2 protein turnover and the potential role of Arl2 in regulating Cdk5rap2 protein levels in mNPCs. The centrosome and the primary cilium at the apical radial glial cells are intricately connected, both of which control NPC proliferation [47]. Interestingly, a recent study showed a role of Arl2 in cilia stability in rod photoreceptor neurons, as Arl2Q70L overexpression caused decreased function and degeneration of these cells [42]. Future study is warranted to determine whether Arl2 is also involved in ciliogenesis in radial glial cells.

Radial glial cells exhibit a bipolar morphology with an apical process anchored to the ventricular surface and a basal process projecting towards the pial surface of the brain [48]. The centrosomes are located at the apical endfoot of the apical process, while microtubules in the basal process are largely acentrosomal where γ-tubulin was undetectable and instead are organized by Golgi outposts [49]. Whether Arl2 can also potentially involved in microtubule assembly within the basal process remains unknown and will be intriguing for future investigations.

Loss-of-function variants of Cdk5rap2 are associated with primary autosomal-recessive microcephaly (MCPH) [50]. Although Arl2 variants have not been found in brain disorders, recent studies identified Arl2 as a candidate gene for an eye disorder [29] and its role in early photoreceptor development via its microtubule functions [51]. Interestingly, mouse Cdk5rap2 was also shown recently to be required for eye development by affecting retina progenitor cell proliferation and apoptosis [52], suggesting that Cdk5rap2 might be linked to Arl2 in other cell types beyond the developing cortex.

Taken together, our study highlights the critical role of Arl2 regulates NPC proliferation and neuronal migration during mouse cortical development. Mechanistically, Arl2 physically associates and recruits Cdk5rap2 to the centrosomes to promote microtubule assembly in NPCs and neuronal migration. These discoveries may facilitate the development of potential therapeutic strategies for neurodevelopmental disorders.

## Materials and methods

### Animals

All animal studies were performed under the Institutional Animal care and use committee (IACUC) approved protocol (IACUC Protocol: 2016/SHS/1207 and 2021/SHS/1672). C57BL/6 mice were purchased from InVivos for the IUE and for primary mNPC culture experiments.

### DNA constructs

Arl2 full-length cDNA and 3 mutant form (Arl2$^{Q70L}$, Arl2$^{T30N}$, and Arl2$^{K71R}$) and Cdk5rap2 from mouse and human were cloned into FUGW (Addgene plasmid # 14883) [53], FUtdTW (Addgene plasmid # 22478) [54], pBiFC-VC155 (Addgene plasmid # 22011) [55], and pBiFC-VN155 (I152L) (Addgene plasmid # 27097) [56] constructs. Small hairpin RNAs were cloned into pGreenPuro constructs from SBI, System Biosciences (cat no: #s SI505A-1). Two shRNAs target different regions of mouse Arl2 (shArl2-1 and shArl2-2), Cdk5rap2 (shCdk5rap2-1 and shCdk5rap2-2), and 1 control shRNA with scrambled sequence were designed.

The following different sets of short hairpin sequences were cloned into pGreenPuro vectors: shArl2-1 (CATCGACTGGCTCCTTGATGACATTTCCA) and shArl2-2 (GACACTGG GCTTCAACATCAAGACCCTGG); shCdk5rap2-1 (GCACATCTACAAGACGAACAT) (Sigma, TRCN0000179786) and shCdk5rap2-2 (GCCATCAAGATACGATTCATT) (Sigma, TRCN0000183538).

### HEK293T culture and lentiviruses package

Clontech's HEK 293T cell line were cultured in D-MEM high glucose medium (Invitrogen), containing 4.5 g/L D-glucose, and 4 mM L-glutamine. For packaging viral vector, high titers of engineered lentiviruses were produced by cotransfection of lentiviral vectors (FUGW, or FUtdTW or pGreenPuro), psPAX2 and pMD2.G into HEK293T cells followed by ultracentrifugation of viral supernatant as previously described [57].

### Mouse neural progenitor cells (mNPCs) culture

Mouse embryos were harvested at E14, and the dorsolateral cortex was dissected and enzymatically triturated to isolate NPCs (S1A Fig). NPCs were suspension-cultured in Costar 6-well Clear Flat Bottom Ultra-Low Attachment Multiple Well Plates (Corning) in proliferation medium (NeuroCult Proliferation Kit (Mouse & Rat), STEMCELL) containing human EGF (10 ng/ml), human FGF2 (10 ng/ml) (Invitrogen, Carlsbad, CA), N2 supplement (1%) (GIBCO), penicillin (100 U/ml), streptomycin (100 mg/ml), and L-glutamine (2 mM) for 7 days and were allowed to proliferate to form neurospheres. DIV 7 neurospheres were dissociated into single cells using accutase, yielding 5 to $6 \times 10^6$ cells per 6-well plate. For proliferation assay, 48-h lentivirus (pPurGreen) infection, the cells were pulsed with 1 mM EdU (Invitrogen) for 3 h. In vitro NPC differentiation assay, mNPC cells were seeded onto 24-well plate with 60 mm coverslips coated with poly-L-lysine, at a density of $4.5 \times 10^4$ cells/coverslip. Twenty-four hours lentivirus (pPurGreen) infection, NPCs were cultured as monolayer in

differentiation medium containing B27 (2%) in Neurobasal medium and were maintained for 5 to 6 days.

## Cortical primary neuron culture

Primary cultures of cortical neurons were prepared from embryonic day 18 (E18) mice as previously described [57]. Briefly, the cortex was carefully dissected from the E18 brain in Earle's Balanced Salt Solution (EBSS, Gibco 0766) and collected in buffer (127 mM NaCl, 5 mM KCl, 170 μM Na2HPO4, 205 μM KH2PO4, 5 mM glucose, 59 mM sucrose, 100 U/mL penicillin/streptomycin (pH 7.4)). Cells were dissociated using 25 mg/ml papain. After collection in growth medium (Dulbecco's Modified Eagle Medium w/GlutaMax (Invitrogen) containing 1 M HEPES, 10% heat-inactivated horse serum (Invitrogen), and 100 U/mL penicillin/dtreptomycin (pH 7.4)) cells were filtered through a 70-μM cell strainer. Subsequently, cells were seeded onto 24-well plate with 60 mm coverslips coated with poly-L-lysine, at a density of $4.5 \times 10^4$ cells/coverslip.

## In utero electroporation

IUE was performed as described previously [58]. Pregnant E13 mice were anesthetized with isoflurane and proceeded with the laparotomy procedure. Small hairpin plasmid DNA with the GFP or overexpression plasmid with tdTomato reporter (2 to 3 μg/μl) was injected into the lateral ventricles of the embryos through the uterine wall. Subsequently, for the electroporation, 4 electrical pulses of 35V, 50 ms was administered with the electroporator device, and the mice were allowed to undergo normal development after the surgery. The electroporated embryonic mice brains were harvested at E14, E15, E16, and E17 for the cell proliferation, differentiation, and migration analysis. Each N value (1 embryo) used for quantification comes from a different litter.

## EdU (5-Ethynyl-2′-deoxyuridine) incorporation assay

For EdU labelling experiments in mice, EdU was injected intraperitonially into the pregnant mice and the mice were killed after 6 h for brain harvest. The brain samples were subjected to standard immunochemistry procedure. The incorporated EdU was detected using fluro azide from ClickiT EdU Imaging Kit (Invitrogen).

## Tissue preparation and immunostaining analysis

Embryonic mice were dissected in phosphate buffered saline (PBS), and the embryonic brain samples were fixed in 4% paraformaldehyde overnight; subsequently, the brain samples were stored in 30% sucrose prior to sectioning. The brain samples were mounted in Tissue-Tek embedding medium and were sectioned using cryostat. For mNPCs, the cells were grown on coverslips and fixed with 4% paraformaldehyde for 15 min at RT. Subsequently, the cells were washed with PBS twice and stored prior to staining. Cells and brain sections were consequently washed with TBS and blocked with 5% normal donkey serum in TBS with 0.1% Triton X (TBST). Respective primary antibodies were prepared with the blocking solution and incubated for 2 h at RT. Subsequently, the cells were washed with TBST and proceeded with secondary fluorophore antibodies incubation for 1 h at RT. For the detection of EdU-incorporated cells and tissues, Alexa Fluor azide was used as per the protocol described (ClickiT EdU Imaging Kits; Invitrogen). The cells and tissues were washed and mounted for imaging. Micrographs were taken using LSM710 confocal microscope system (Axio Observer Z1; ZEISS), fitted with a PlanApochromat 40×/1.3 NA oil differential interference contrast objective, and brightness and contrast were adjusted by ImageJ.

The primary antibodies used in this paper, rabbit anti-TBR1 (1:500; Cell Signalling, cat no: 49661S), rabbit anti-TBR2 (1:500; Abcam, cat no: ab23345), rabbit anti-Pax6 (1:500; BioLegend, cat no: B328397), mouse anti-alpha tubulin (1:1,000; Sigma, cat no: T6199), mouse anti-gamma tubulin (1:500; Sigma, cat no: T5326), mouse anti-DCX (1:300; Cruz Biotechnology, cat no: A0919), rabbit anti-Ki67 (1:500; Abcam, cat no: ab16667), rat anti-PH3 (1:500; sigma, cat no: 4882), rabbit anti-NeuroD2 (1:500, abcam, cat no: ab104430), rabbit anti-caspase3 (1:500; BD Pharmingen, cat no: 559569), rabbit anti-Cdk5rap2 (1:500, Merck Millipore, cat no: 06–1398), rabbit anti-Arl2 (1:300; Abcam, cat no: ab183510), mouse anti-myc (1:500; Abcam, cat no:1011022–5), guinea pig anti-GFP (1:1,000; Dr. Yu Feng Wei lab), mouse anti-GFP (1:1,000; Dr. Yu Feng Wei lab), rabbit anti-HA (1:500; Sigma, cat no: H6908), rat anti-HA (1:500; Roche, cat no:423001).

## Proximity ligation assay

PLA was performed as described ([59]; adopted from Duolink PLA, Merck). mNPCs were transfected with the following constructs including control-GFP, control-myc, Arl2-GFP, and Cdk5rap2-myc using Lipofectamine Transfection reagent (Invitrogen). The cells were washed with cold PBS thrice and fixed with 4% paraformaldehyde in PB for 15 min. Subsequently, the cells were blocked with 5% normal donkey serum in TBS-Tx (0.1% Triton-X100) for 45 min. The cells were incubated with respective primary antibodies at RT for 2 h. The cells were then incubated with PLA probes at 37°C for 1 h. Subsequently, the cells were washed Buffer A for 5 min at RT. The cells were proceeded with ligation of probes at 37°C for 30 min and amplification at 37°C for 1.5 h, followed by 2 washes with Buffer B at RT. The cells were washed once with 0.01× Buffer B and proceeded with primary antibodies incubation diluted in 3% bovine serum albumin (BSA) in PBS for 2 h at RT. Following this, the cells were washed twice with 0.1% TBS-TX and incubated with secondary antibodies for 1.5 h at RT. The cells were subsequently washed with PB and then mounted using in situ mounting media with DAPI (Duolink, Sigma-Aldrich).

## Microtubule regrowth assay

mNPCs were incubated with 2.5 mM thymidine at 37°C for 20 h to induce S phase arrest (Fig 3A). Subsequently, the cells were released from S phase arrest for 7 h. The cells were then incubated with 200 nM nocodazole for 4 h. The cells were washed with ice-cold medium and incubated on ice for 30 min to initiate microtubule depolymerization. The cells were subsequently replaced with prewarmed medium at 37°C. The cells were washed with PBS and were incubated in 4% paraformaldehyde for 15 min to fix the cells. The standard immunochemistry assay was performed to quantify the Alpha-tubulin density.

## Co-immunoprecipitation

Cells were lysed using PierceTM IP lysis buffer (Thermo Fisher Scientific, cat no: 87787) with protease inhibitors. For input controls, 1% cell lysate was taken, and the remaining were incubated with respective pulldown antibodies overnight at 4°C. Protein A/G ultralink resin beads (Thermo Fisher Scientific, cat no: 53135) were added to the cell lysate and incubated for 3 h at 4°C. Consequently, the beads were washed with PBS for several times to remove the residual proteins. The beads were mixed with the protein loading dye and proceeded for WB analysis.

## Western blot analysis

Cells were lysed using PierceTM RIPA buffer (Thermo Fisher Scientific, cat no: 89901) with protease inhibitors. The proteins samples were separated using SDS-PAGE and were

transferred onto the nitrocellulose membrane. The membranes were blocked with low-fat dry milk in PBS with 0.1% Tween20 (PBST) for 1 h at RT. Subsequently, the membranes were incubated with respective primary antibodies in 5% BSA with PBST overnight at 4˚C. The membranes were washed thrice with PBST and incubated with HRP-conjugated secondary antibodies to probe the target proteins for 1 h at RT. The membranes were washed and the proteins were detected using SuperSignal West Pico Chemiluminescence Substrate (Protein Biology, cat no: 34580).

## AlphaFold2 multimer protein complex prediction

To discover Arl2 interactors with centrosome proteins, we performed protein complex predictions using Alphafold multimer developed by DeepMind [60]. Arl2 was predicted against core centrosome proteins. All the predictions were performed using AlphaPulldown Pipeline v0.30.6 [37] with default settings. Multiple sequence alignments (MSAs) and template input to the Alphafold multimer were calculated by MMseqs2 [38]. To analyze the results produced by AlphaFold multimer, ipTM scores from the predictions were used to evaluate the interaction possibility and confidence. Predicted interaction structure models were used for further analysis.

## Spinning disk super-resolution imaging

Super-resolution imaging was performed as previously described [61]. In brief, super-resolution spinning disk confocal-structured illumination microscopy (SDC-SIM) was performed on a spinning disk system (Gataca Systems) based on an inverted microscope (Nikon Ti2-E; Nikon) equipped with a confocal spinning head (CSU-W; Yokogawa), a Plan-Apo objective (100× 1.45-NA), a back-illuminated sCMOS camera (Prime95B; Teledyne Photometrics), and a super-resolution module (Live-SR; GATACA Systems). All image acquisition and processing were controlled by the MetaMorph (Molecular Device) software. Images were further processed with ImageJ.

## Live-cell imaging

To capture time-lapse images of mNPCs, a super-resolution SDC-SIM equipped with a Plan-Apo objective (100× 1.45-NA) was used. The imaging was conducted in a chamber at a temperature of 37˚C with $CO_2$ supplement. mNPCs were imaged for 16 h (5 min each time interval for Figs 4H and S5A) or for 5 min without intervals (for Fig 5C). The videos were processed using an ImageJ software. Mito-RFP tracker (Plasmid #51013) was from Addgene (pLenti.CAG.H2B-cerFP-2A-mito-dsRFP.W). Viafluor-488 live cell microtubule staining kit (Biotium, #70062) and Viafluor-647 live cell microtubule staining kit (Biotium, #70063) were used for live imaging of mNPCs in vitro.

## Tracking of EB3-GFP or EB3-Td comets

mNPCs expressing EB3-GFP or EB3-Td were subjected to live-cell imaging using a super-resolution SDC-SIM as mentioned above. The amount and velocity of the EB3-GFP comets were calculated, and kymographs were generated using KymoButler [62]. A cell was imaged for 3 to 5 min without time interval for each movie, and videos were generated with NIH ImageJ software.

## Statistical analysis

All experiments were repeated at least thrice, and comparable results were obtained. All statistical analysis was performed using GraphPad prism. Paired or unpaired $t$ test were used for the

comparison of 2 independent groups, and one-way or two-way ANOVA were used for comparing more than 2 independent groups. Statistical significance was represented by ***$p < 0.001$, **$p < 0.01$, *$p < 0.05$ compared with the control groups.

### Analyzing single-cell RNA sequencing dataset

Mouse embryonic brain scRNA-seq dataset with cell-type annotations from La Manno and colleagues [31] were retrieved. Seurat v5 and R 4.4.0 were used to examine Arl2 expression level among all the neuronal cells and glial cells. Sox2, an NSC marker, was included in the analysis as a control. Nonneuronal and nonglial cells, such as fibroblasts, endoderm, and ectodermal cells, were excluded from this analysis.

## Supporting information

**S1 Fig. Arl2 KD affects mNPC proliferation in vitro.** (**A**) WB was performed to detect Arl2 protein in mouse cerebral cortical tissue isolated from E12, 13, 14, 16, P0, P7. (**B**) Bar graphs showing Arl2 expression levels at E12, $1.00 \pm 0.00$ A.U.; E13, $0.91 \pm 0.10$ A.U.; E14, $0.57 \pm 0.17$ A.U.; E16, $0.41 \pm 0.16$ A.U.; P1, $0.24 \pm 0.09$ A.U.; P7, $0.18 \pm 0.06$ A.U. $n = 3$. (**C**) Arl2 has the highest expression in radial glial cells in mouse developing brain. The expression of Arl2, along with Sox2, an NSC marker, was reanalyzed in the single-cell RNA sequencing dataset that profiles the mouse embryonic brain each day between E7-18 [31]. (**D**) Schematic representation of primary mNPC culture from embryonic mice. (**E**) WB analysis of mNPC protein extracts of control (H1-shCtrl-GFP) and Arl2 KD (H1-shArl2-GFP) with lentivirus (pPur-Green) infection in 48-h culture. Blots were probed with anti-Arl2 antibody and anti-GAPDH antibody. Immunoblot showing the efficiency of Arl2 KD. (**F**) Bar graphs representing KD efficiency of Arl2 normalized to the internal control GAPDH ($0.68 \pm 0.28$ in shArl2-1, $0.33 \pm 0.05$ in shArl2-2 vs. $1.60 \pm 0.15$ in shCtrl, $n = 3$). (**G**) Immunostaining micrographs of mNPCs labelled for EdU and the cell apoptosis marker active Caspase-3. (**H**) Bar graphs showing reduced EdU incorporation upon Arl2 KD ($34.35 \pm 5.95\%$ in shArl2 with 12 images vs. $46.03 \pm 0.95\%$ in shCtrl with 14 images, $n = 3$ batches). (**I**) Bar graphs showing increased Caspase-3+ cells in shArl2 group compared to the control ($25.86 \pm 3.26\%$ in shArl2 vs. $9.64 \pm 0.86\%$ in shCtrl, $n = 3$ batches). (**J**) Time series showing decreased cell proliferation upon Arl2 KD as compared to control. (**K**) Line graph representing the timeline of mNPC proliferation and showing the defects in cell proliferation in shArl2 group compared to the control. The values represent the mean ± SD. One-way ANOVA in C. Student $t$ test in E, F. Differences were considered significant at at *$p < 0.05$, **$p < 0.01$, and ****$p < 0.0001$. Scale bars; F and I = 150 μm. Source data can be found in S1 Data. Arl2, ADP ribosylation factor-like GTPase 2; EdU, 5-ethynyl-2′-deoxyuridine; KD, knockdown; mNPC, mouse neural progenitor cell; WB, western blot.
(TIF)

**S2 Fig. Arl2 KD affects mNPC proliferation in vivo.** (**A**) Immunostaining micrographs showing decreased neural complexities of in primary cortical neurons (labeled by Map2) in vitro in shCtrl and shArl2. (B) An illustration of Sholl's analysis of neurite morphology. (**C**) Sholl's analysis showing the distance from soma and the intersection number reduced in shArl2 as compared to the control. (**D**) Sholl's analysis showing the total intersection number as measured and significantly reduced in shArl2 ($22.78 \pm 5.04$) as compared to control ($64.17 \pm 2.62$). (**E**) Brain slices from shCtrl (scrambled control) and shArl2 (Arl2 shRNA) groups at E14, 1 day after IUE, were labelled with active Caspase-3 and GFP. (**F**) Bar graph showing increased caspase-3+/GFP+ cells in the VZ and SVZ upon Arl2 KD ($15.7 \pm 1.85\%$) as

compared to control (2.65 ± 0.80%). The values represent the mean ± SD ($n$ = X embryos). Student $t$ test; differences were considered significant at ****$p$ < 0.0001. Source data can be found in S1 Data. Arl2, ADP ribosylation factor-like GTPase 2; GFP, green fluorescent protein; IUE, in utero electroporation; KD, knockdown; mNPC, mouse neural progenitor cell; shRNA, short hairpin RNA; SVZ, subventricular zone; VZ, ventricular zone.
(TIF)

**S3 Fig. Overexpression Arl2 and mutant affect mNPC proliferation.** (**A**) Immunostaining micrographs of mNPCs in vitro in Arl2$^{WT}$, Arl2$^{T30N}$, and Arl2$^{Q70L}$ labelled with EdU and caspase-3. (**B**) Bar graphs showing the proportion of EdU+ cells in control (59.06 ± 8.52%), Arl2$^{WT}$ (79.01 ± 5.34%), Arl2$^{T30N}$ and Arl2$^{Q70L}$ (36.4 ± 4.93% and 27.4 ± 5.78%, respectively) ($n$ = 3). (**C**) Bar graphs showing the proportion of caspase-3 + cells in control (22.18 ± 9.21%, $n$ = 4), Arl2$^{WT}$ (30.89 ± 3.22%, $n$ = 4), Arl2$^{T30N}$ and Arl2$^{Q70L}$ (86.76 ± 6.36% and 71.86 ± 6.74%, respectively) ($n$ = 3). (**D**) Brain slices from Arl2$^{WT}$, Arl2$^{T30N}$, and Arl2$^{Q70L}$, at E15, 2 days after IUE, labelled for phospho-histone H3-positive (PH3+). (**E**) Bar graphs showing the proportion of PH3+ cells in control 3.31 ± 1.56%, $n$ = 4; Arl2$^{WT}$ 2.94 ± 0.53%, $n$ = 4; Arl2$^{T30N}$ 8.31 ± 2.08% and Arl2$^{Q70L}$ 9.42 ± 1.78%, $n$ = 4, in 2 days after IUE in VZ of brain sections. The values represent the mean ± SD. One-way ANOVA with multiple comparisons in B, C, and E. Differences were considered significant at *$p$ < 0.05, **$p$ < 0.01, ***$p$ < 0.001, and ****$p$ < 0.0001, ns = nonsignificance. Scale bars; A and D = 50 μm. Source data can be found in S1 Data. Arl2, ADP ribosylation factor-like GTPase 2; EdU, 5-ethynyl-2′-deoxyuridine; IUE, in utero electroporation; mNPC, mouse neural progenitor cell; VZ, ventricular zone.
(TIF)

**S4 Fig. Overexpression Arl2 mutant affect mNPC proliferation.** (**A**) Immunostaining micrographs of mNPCs in vitro in Arl2$^{WT}$, Arl2$^{T30N}$, and Arl2$^{Q70L}$ labelled for phospho-histone H3-positive (PH3+). (**B**) Brain slices from Arl2$^{WT}$ and Arl2$^{Q70L}$, at E16, 3 days after IUE, labelled for caspase-3, tdTomato, and PH3. (**C**) Bar graph showing the caspase-3 staining and the proportion of caspase-3 + cells in the IZ in Arl2$^{Q70L}$ (37.28 ± 3.56%, $n$ = 5) as compared to control (4.45 ± 3.12%, $n$ = 4) in 3 days after IUE. (**D**) Brain slices from Vector control, Arl2$^{WT}$ and Arl2$^{Q70L}$, at E16, 3 days after IUE, labelled for Neuro-D, tdTomato, and Hoechst. (**E**) Bar graph showing the expression of NeuroD2, a neuronal marker found in immature neurons, in Arl2$^{WT}$ (30.77 ± 2.93%), but dramatically reduced in Arl2$^{Q70L}$ (6.75 ± 2.69%) 3 days after IUE as compared to control (20.57 ± 1.36%). The values represent the mean ± SD. One-way ANOVA in E. Student $t$ test in C; differences were considered significant at **$p$ < 0.01, ***$p$ < 0.001, and ****$p$ < 0.0001, ns = nonsignificance. Scale bars; A = 5 μm; B = 50 μm; C = 100 μm. Source data can be found in S1 Data. Arl2, ADP ribosylation factor-like GTPase 2; IUE, in utero electroporation; IZ, intermediate zone; mNPC, mouse neural progenitor cell.
(TIF)

**S5 Fig. Arl2-K71R mutant causes mitochondrial fragmentation and has no effect on microtubule regrowth.** (**A**) Immunostaining micrographs showing decreased neural complexities of primary cortical neurons (labeled by Map2) in vitro in overexpression of Arl2$^{T30N}$ and Arl2$^{Q70L}$ mutants. (**B**) Sholl's analysis showing the distance from soma and the intersection number reduced in the ARL2$^{T30N}$ and ARL2$^{Q70L}$ mutants as compared to the control. (**C**) Sholl's analysis showing the total intersection number as measured and significantly reduced in Arl2$^{Q70L}$ (11 ± 1.80) and Arl2$^{T30N}$ (8.83 ± 1.04) as compared to control (38 ± 4.58). (**D**) Immunostaining micrographs of mNPCs in Vector control and mouse Arl2$^{K71R}$ overexpression. Arl2$^{K71R}$ overexpression showed fragmented mitochondria with shortened mitochondrial length as compared to control in mNPCs in vitro. (**E**) Graph representing mitochondrial

length (2.41 ± 0.99 μm) in Arl2$^{K71R}$ mutant as compared to control (5.53 ± 3.78 μm). (**F**) Cortical brain sections following overexpression of mouse Arl2$^{WT}$ (mArl2$^{WT}$) and mouse Arl2$^{K71R}$ at E16, 3 days after IUE, were labelled with tdTomato (Td). (**G**) Bar graphs (images in F) representing Td+ cell population in the group of mouse Arl2$^{WT}$ (mArl2$^{WT}$) (VZ = 14.80 ± 1.84%, SVZ = 18.57 ± 2.28%, IZ = 35.32 ± 1.05%, CP = 31.30 ± 2.15%, $n$ = 4 embryos) and Arl2$^{K71R}$ mutant (VZ = 12.54 ± 3.09%, SVZ = 17.44 ± 2.59%, IZ = 37.31 ± 2.36%, CP = 31.30 ± 2.15%, $n$ = 5 embryos). The values represent the mean ± SD. One-way ANOVA in C. Student $t$ test in E. Multiple $t$ test in G, $n$ = 3. Differences were considered significant at $^{**}p < 0.01$ and $^{****}p < 0.0001$, ns = nonsignificance. Scale bars; A = 20 μm; D = 5 μm; F = 100 μm. Source data can be found in S1 Data. CP, cortical plate; IUE, in utero electroporation; IZ, intermediate zone; mNPC, mouse neural progenitor cell; SVZ, subventricular zone; VZ, ventricular zone. (TIF)

**S6 Fig. Mitochondrial morphology remained unchanged upon shArl2 KD or overexpression of Arl2 in mouse brains.** (**A**) Live imaging micrograph of mitochondria (Mito-RFP, Plasmid #51013, Addgene) and microtubule (Viafluor-488 live cell microtubule staining kit (Biotium, #70062) dynamics in control, Arl2$^{Q70L}$-overexpressing, or Arl2$^{T30N}$-overexpressing mNPCs. (**B**) Bar graph shows qualifications of mitochondria morphology in various genotypes in A. (**C**) Brain slices from shCtrl, Arl2$^{WT}$, and shArl2 groups at E17, 4 days after IUE, were labelled with GFP, TdTomato, Mito, and DNA. Scale bars; A and C = 10 μm. Source data can be found in S1 Data. Arl2, ADP ribosylation factor-like GTPase 2; GFP, green fluorescent protein; IUE, in utero electroporation; KD, knockdown; mNPC, mouse neural progenitor cell. (TIF)

**S7 Fig. The phenotypes of Cdk5rap2 KD are similar with Arl2 KD.** (**A**) WB analysis of mNPC protein extracts of control (H1-shCtrl-GFP) and Cdk5rap2 KD (H1-shCdk5rap2-GFP) with lentivirus (pPurGreen) infection in 48-h culture. Blots were probed with anti-Cdk5rap2 antibody and anti-GAPDH antibody. (**B**) Bar graphs representing KD efficiency of Cdk5rap2 normalized to the internal control GAPDH (0.26 ± 0.21 in shCdk5rap2-1, 0.32 ± 0.20 in shCdk5rap2-2 normalized in shCtrl, $n$ = 3). (**C**) Immunostaining micrographs in shCtrl, shArl2, shCdk5rap2-1, and shCdk5rap2-2 in interphase mNPCs were labelled with γ-tubulin, Cdk5rap2, and Hoechst (DNA). (**D**) Quantification graph showing Cdk5rap2 intensity at interphase of mNPCs (46.14 ± 12.51) upon Arl2 KD and (shCdk5rap2-1 = 34.73 ± 15.93; shCdk5rap2-2 = 44.53 ± 13.16) upon Cdk5rap2 KD as compared to shCtrl (77.37 ± 29.09, $n$ = 3 batches with 20 cells). (**E**) Quantification graph showing γ-tubulin intensity at interphase of mNPCs (52.83 ± 10.22) upon Arl2 KD and (shCdk5rap2-1 = 32.02 ± 7.45; shCdk5rap2-2 = 36.29 ± 13.21) upon Cdk5rap2 KD as compared to shCtrl (80.47 ± 23.81, $n$ = 3 batches with 10 cells). The values represent the mean ± SD. One-way ANOVA in D and E. Differences were considered significant at $^*p < 0.05$, $^{**}p < 0.01$, and $^{****}p < 0.0001$, ns = nonsignificance. (**F**) Immunostaining micrographs in shCtrl, shCdk5rap2-1, and shCdk5rap2-2 in mNPCs were labelled with EdU and DNA. (**G**) Bar graphs showing reduced EdU incorporation upon Cdk5rap2 KD in mNPCs. The values represent the mean ± SD (shCtrl = 58.84 ± 6.06%; shCdk5rap2-1 = 26.42 ± 7.89%; shCdk5rap2-2 = 34.58 ± 5.72%, $n$ = 3). (**H**) Brain slices from shCtrl, shCdk5rap2 groups at E17, 4 days after IUE, were labelled with GFP. (**I**) Box plots representing GFP+ cells for (H) in CP (shCtrl: 81.51 ± 3.26%, shCdk5rap2: 58.80 ± 2.17%), IZ (shCtrl: 14.29 ± 2.86%, shCdk5rap2: 27.40 ± 2.62%), and SVZ + VZ (shCtrl: 4.20 ± 2.73%, shCdk5rap2: 13.80 ± 2.85%) (shCtrl: $n$ = 4, shCdk5rap2: $n$ = 4) showing defects in neuronal migration to CP upon Cdk5rap2 KD compared to the control. The values represent the mean ± SD. One-way ANOVA in B, D, E, and G. Multiple unpaired $t$ tests in I. Differences were considered significant at $^{**}p < 0.01$, $^{***}p < 0.001$, and $^{****}p < 0.0001$.

ns = nonsignificance. Scale bars; C = 5 μm; F = 50 μm; H = 80 μm. Source data can be found in S1 Data. Arl2, ADP ribosylation factor-like GTPase 2; Cdk5rap2, CDK5 Regulatory Subunit Associated Protein 2; CP, cortical plate; EdU, 5-ethynyl-2′-deoxyuridine; GFP, green fluorescent protein; IUE, in utero electroporation; IZ, intermediate zone; KD, knockdown; mNPC, mouse neural progenitor cell; SVZ, subventricular zone; VZ, ventricular zone; WB, western blot.
(TIF)

**S1 Movie. mNPCs proliferation in shCtrl by Incucyte in S1G Fig.** Time-lapse imaging to track the proliferation of mNPCs by Incucyte in control group for 4 days. Time scale: day: hour: minute. Scale bar: 1.10 mm. (AVI)
(AVI)

**S2 Movie. mNPCs proliferation in shArl2 by Incucyte in S1G Fig.** Time-lapse imaging to track the proliferation of mNPCs by Incucyte in Arl2 KD group for 4 days. Time scale: day: hour: minute. Scale bar: 1.1 mm. (AVI)
(AVI)

**S3 Movie. Arl2-KD in mNPCs for mitotic duration in Fig 4A.** Live imaging of mNPCs in vitro using the Viafluor-647 live cell microtubule staining kit (Biotium, #70063) in mNPCs in both shCtrl and shArl2 groups. Time scale: minute: second. Scale bar: 10 μm. (AVI)
(AVI)

**S4 Movie. Arl2-OE in mNPCs for mitotic duration in Fig 4C.** Live imaging of mNPCs in vitro using the Viafluor-488 live cell microtubule staining kit (Biotium, #70062) in mNPCs overexpressing Arl2$^{WT}$, Arl2$^{Q70L}$, and Arl2$^{T30N}$. Time scale: hour: minute. Scale bar: 10 μm. (AVI)
(AVI)

**S5 Movie. Arl2-KD with EB3-Td in mNPCs in Fig 5D.** Time-lapse imaging to track the growing ends of microtubules by using the plus-end microtubule binding protein EB3 tagged with Tdtomato (Td) in mNPCs in both shCtrl and shArl2 groups. Time scale: minute: second. Scale bar: 10 μm. (AVI)
(AVI)

**S6 Movie. Arl2-OE with mitochondria dynamics in mNPCs in S6A Fig.** Time-lapse imaging showing the mitochondrial morphology and microtubules in the overexpression of ARL2$^{Q70L}$ and ARL2$^{T30N}$ mutants. Time scale: minute: second. Scale bar: 10 μm. (AVI)
(AVI)

**S7 Movie. Arl2-OE with EB3-GFP in mNPCs in Fig 6D.** Time-lapse imaging to track the EB3 tagged with GFP in mNPCs in overexpression of Arl2$^{WT}$, Arl2$^{Q70L}$, and Arl2$^{T30N}$. Time scale: minute: second. Scale bar: 10 μm. (AVI)
(AVI)

**S1 Data. Excel file with all individual numerical values corresponding to the data presented in the main and supporting figures.** Corresponding figure numbers are indicated in each Excel worksheet.
(XLSX)

**S1 Raw Images. PDF file with all raw images of WB corresponding to the data presented in the main and supporting figures.**
(PDF)

## Author Contributions

**Conceptualization:** Dongliang Ma, Hongyan Wang.

**Data curation:** Dongliang Ma, Kun-Yang Lin, Divya Suresh, Jiaen Lin, Mahekta R. Gujar, Htet Yamin Aung, Ye Sing Tan, Yang Gao, Anselm S. Vincent, Teng Chen.

**Formal analysis:** Dongliang Ma, Kun-Yang Lin, Divya Suresh, Jiaen Lin, Mahekta R. Gujar, Htet Yamin Aung, Ye Sing Tan, Yang Gao, Anselm S. Vincent, Teng Chen.

**Funding acquisition:** Hongyan Wang.

**Investigation:** Dongliang Ma.

**Methodology:** Dongliang Ma, Kun-Yang Lin, Divya Suresh, Jiaen Lin, Mahekta R. Gujar, Htet Yamin Aung, Ye Sing Tan, Yang Gao, Anselm S. Vincent, Teng Chen.

**Resources:** Hongyan Wang.

**Software:** Dongliang Ma.

**Supervision:** Hongyan Wang.

**Validation:** Dongliang Ma.

**Visualization:** Dongliang Ma.

**Writing – original draft:** Dongliang Ma, Mahekta R. Gujar, Hongyan Wang.

**Writing – review & editing:** Dongliang Ma, Mahekta R. Gujar, Hongyan Wang.

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
