## [Editor Report · Decision Letter 0]

11 Jan 2024

Dear Dr Wang, 

Thank you for submitting your manuscript entitled "Arl2 Associates with Cdk5rap2 to Regulate Cortical Development via Microtubule Organization" for consideration as a Research Article by PLOS Biology.

Your manuscript has now been evaluated by the PLOS Biology editorial staff as well as by an academic editor with relevant expertise and I am writing to let you know that we would like to send your submission out for external peer review.

Once your full submission is complete, your paper will undergo a series of checks in preparation for peer review. After your manuscript has passed the checks it will be sent out for review. To provide the metadata for your submission, please Login to Editorial Manager (https://www.editorialmanager.com/pbiology) within two working days, i.e. by Jan 13 2024 11:59PM.

Kind regards,

Christian

Christian Schnell, PhD

Senior Editor

PLOS Biology

cschnell@plos.org

---

## [Decision Letter · Decision Letter 1]

26 Feb 2024

Dear Dr Wang,

Thank you for your patience while your manuscript "Arl2 Associates with Cdk5rap2 to Regulate Cortical Development via Microtubule Organization" was peer-reviewed at PLOS Biology. It has now been evaluated by the PLOS Biology editors, an Academic Editor with relevant expertise, and by several independent reviewers. 

In light of the reviews, which you will find at the end of this email, we would like to invite you to revise the work to thoroughly address the reviewers' reports.

As you will see below, the reviewers reviewers think that the study is very well executed and provides important insights. However, they raise some concerns regarding lack of control experiments and small sample sizes, which we would like you to address. Please note that after discussing the reviewers' comments with the Academic Editor, we think that you can remove the “asymmetric vs symmetric divisions" part from the manuscript, to focus on the differentiation outcome and migration effects.

Given the extent of revision needed, we cannot make a decision about publication until we have seen the revised manuscript and your response to the reviewers' comments. Your revised manuscript is likely to be sent for further evaluation by all or a subset of the reviewers.

**IMPORTANT - SUBMITTING YOUR REVISION**

*Re-submission Checklist*

*Published Peer Review*

*PLOS Data Policy*

*Blot and Gel Data Policy*

Sincerely,

Christian

Christian Schnell, PhD

Senior Editor

PLOS Biology

cschnell@plos.org

REVIEWS:

Reviewer #1: This manuscript by Dong-Lian Ma focuses on the role of Arl2 (ADP ribosylation factor-like GTPase 2) in mouse cortical neurogenesis and its molecular interactions. The authors, who have discovered Arl2 is involved neural progenitor cell divisions, found that this is also the case. They discovered that mouse Arl2 plays a new role in corticogenesis via regulating microtubule growth, but not mitochondria functions, and that it molecularly interacts with centrosomal protein, Cdk5Rrap2, whose loss is known to cause microcephaly. The authors revealed Arl2 functions upstream of Cdk5Rrap2. Thus, the authors propose that Arl2 plays critical role in mammalian corticogenesis via controlling centrosome and/or microtubule functions.

The manuscript includes fascinating findings regarding molecular and cell biological functions of Arl2 in mammalian brain development and microtubule dynamics that are required for cell migration and divisions of neural stem cell (NSC)s. The reviewer points out a couple of points that are important and necessary to be improved to be published in PLOS Biology.

Line 119-127

Arl2 KD certainly causes retention of NSCs. The authors indicate that reduced proliferation of NCS in the VZ and SVZ as well as the inhibition of cell migration into the cortical plate. However, no clear test has been done to clarify whether defective migration of VZ/SVZ cells is caused by either "neuron"s migration or defective differentiation from progenitors to "neurons" leading to their retention in the VZ. So it seems to be overstated that Arl2 knock-down (KD) data suggest "neuronal" migration defect in this version of the manuscript.

Line 167-168

The authors state that overexpression of both Arl2T30N and Arl2Q70L results in a similar phenotype in neuronal migration as Arl2 KD. However, the reviwer could not find experiments in this manuscript to examine whether Arl2 KD cause cell cycle arrest (increased PH3+) and/or cell death. The reviewer considers that this is a missing but important data.

Line 169-181

AS for two Arl2 mutants, Arl2[T10N] and arL3 Q70L, the authors clearly observe cell cycle arrest (an increase in PH3+ cells; SupFig.2D in vivo) and extensive cell death in the VZ (SupFig.2A, C, for in vitro, and SupFig.3B for In vivo)

and state "These data suggest that the migration and proliferation defects (mitotic arrest) are observed in Arl2 mutants, which eventually leading to cell death".

The reviewer does not understand why the authors do not test the degree of cell death caused by Arl2KD. These data by using Arl2 KD will bring us much straightforward and consistent interpretation about the results by Arl2 perturbation. 

Line 182-194

A main conclusion of this study is that Arl2 dysfunction results in a defective migration of "neuronal" cells; the authors state that "the expression of NeuroD2, a neuronal marker found in immature neurons, is significantly increased in Arl2WT (30.77 ± 2.93%) but dramatically reduced in Arl2Q70L (6.75 ± 2.69%) 3 days after IUE as compared to control (20.57 ± 1.36%) (Supplementary Figure 3D-E). Taken together, Arl2 dysfunction resulted in a defective migration of neuronal cells."

As far as the reviewer understand correctly, in this study, the authors did not show evidence that defective migration occurs for "differentiated" cells, namely "neuronal cells". The reviewer considers that it would be better to exclude the possibility that the production of neurons is inefficient from progenitors or that immature neurons die before migration. My concern comes from the results regarding NeuroD2+ cells rather looking like that neuronal differentiation are diminished. 

Line 225-241 

The authors claim that loss of Arl2 led to a shift of symmetric division to asymmetric division of mNPCs and alters the mNPC differentiation.

This claim is based on observing perturbed spindle orientations by Arl2 KD, and the authors state that Arl2-depleted NPCs divides asymmetrically frequently than symmetrically as compared to control; line 230 "At E14, one day after IUE, wild-type radial glial cells in the VZ can undergo both symmetric and asymmetric divisions, depending on the plane of division (Figure 4A-C)."

The reviewers did not agree this claim from two points of view. 

First of all, at E14, the stage when layer 4 neurons are generated, most NSCs (RG) divide asymmetrically in their fate (self-renewal vs. differentiation). 

Secondly, it has been established that division plane orientation of cortical NSCs is nothing to do with the division styles self-renewal or differentiation of NSCs (Konno et al. 2008, Shitamukai et al. 2011, Morin & Bellaïche 2011). Rather, cleavage plane orientation is related with the generation of migrating NSCs. If the cleavage plane bypasses the apical membrane, one daughter (without the apical attachment) becomes a migrating self-renewing NSC with another differentiating daughter (with the apical attachment); if the cleavage plane bisects the apical membrane bisects, one daughter becomes a radial glia and the other an differentiating intermediate daughter. 

Given this, it is reasonable that the NSC population in the VZ will not drastically change under the Arl2 KD condition, where a majority of cells (including NSCs) are trapped in the VZ and SVZ.

If the authors want to leep this part in the manuscript, the reviewer also strongly recommends to examine whether the fate of cortical NSCs is precisely followed in the wild type to see whether it depends cleavage orientation as a control and then compare the result with the situation of Arl2 KD. Also, the cleavage plane is defined in the 3D tissue structure. Therefore, it is strongly recommended to take a protocol by which cleavage planes are measured considering the 3D space. 

Reviewer #2: In this study, Ma et al. describe a novel role for Arl2 in mouse cerebral cortex development. The authors, using knockdown and overexpression strategies either in vitro or in vivo, show that Arl2 is important for neural progenitor proliferation and neuronal migration during corticogenesis in mice. These effects are linked to its activity on microtubules via the centrosomal protein Cdk5rap2. The manuscript presents new information that will be of interest to developmental neurobiologists. The experiments are well conducted and convincing. However several points need to be addressed before publication.

Major points:

- The expression pattern of Arl2 in the embryonic cerebral cortex must be described either at the RNA or protein levels on cortical sections. This will help to understand the functions of Arl2 in the developing cortex.

- It is not clear whether the effects on migration are linked to the effect on cell cycle progression. Indeed, a delay in cell cycle progression could explain the reduction of cells reaching the cortical plate at a specific time point. The authors should determine whether Arl2 has a direct role on migration (for example by knocking down Arl2 only on postmitotic neurons or by performing live imaging to visualize migrating neurons). It would help also to look at the morphology of the migrating neurons directly in sections.

-The number of n is sometimes low (n=3). It should be mentioned whether the three embryos used for quantification come from same or different litters.

Minor points:

- The reading is sometimes difficult because the figures are not cited in an ordered manner. For example the mutant Arl2K41R appears on Figure 2 but it is described in the text after figure 4. Similarly, some figure panels are not commented in the text such as Tbr2, Pax6 or Tbr1 stainings in Figure 1.

-The two mutants, dominant negative and dominant active of Arl2, give similar phenotypes. This should be discussed.

Reviewer #3: This manuscript presents a mechanistic link between Arl2 and Cdk5rap2 and neuronal progenitor proliferation and neuronal migration in the developing mouse cortex. The authors have very carefully conducted a thorough set of experiments to support their hypothesis. Using a combination of in utero electroporations, shRNA knockdown, overexpression of Arl2 mutant constructs and primary neural cultures they report a novel finding that Arl2 is important for recruitment of both gamma tubulin and Cdk5rap2 to the centrosome and that the lack of this recruitment leads to increase in mitotic arrest in the Arl2 mutants which is rescued by Cdk5rap2 overexpression. In addition, they showed that Arl2 and Cdk5rap2 interact using multiple methodologies from in silico methods to proximity ligation assays and immunoprecipitation of overexpressed constructs. My comments see below are minor because I consider the authors have done outstanding work on supporting their claims and they provide important insights into the cell biology of neurogenesis.

1- In the abstract they only refer to the interaction between Arl2 and Cdk5rap2 as if it was only done in silico this should be revised to reflect that the authors used additional methods to prove the interaction between these two proteins.

2- Across all figures where there are microscope images I recommend that they put a scale bar in each figure not just in one of the figures in the panel

3- For all figures in which the cortex is being shown I suggest they show an enlarged image for when they want to show colocalization 

4- They have an interesting finding that when Arl2 is knockdown the total levels of Cdk5rap2 are also knockdown they attribute this to the lack of CDk5rap2 localization to the centrosome which leads to a loss of stability? I think it would be important for them to discuss this a little bit more on the discussion because their western blots are from total cell extract. Do they know if they use a protease inhibitor can they restore the levels of Cdk5rap2 even if its localization to the centrosome is still lost? I think that being able to separate these two cellular effects would be important, is it that Arl2 somehow regulates the levels of Cdk5rap2 and that is why there is none in the centrosome or are those two separate from each other. Alternatively a good discussion on this topic would be important to include.

5- For the EB1 studies the P values are out of the total number of measurements or from the comparison of the mean values? There is a lot of overlap between the different groups. In addition, can the authors discuss if their changes in EB1 could be showing changes in microtubule dynamics since this is a marker of growing tips of microtubules

---

## [Editor Report · Decision Letter 2]

11 Jun 2024

Dear Dr Wang,

Thank you for your patience while we considered your revised manuscript "Arl2 Associates with Cdk5rap2 to Regulate Cortical Development via Microtubule Organization" for consideration as a Research Article at PLOS Biology. Your revised study has now been evaluated by the PLOS Biology editors and the Academic Editor.

We are pleased to offer you the opportunity to address the remaining point from the Academic Editor in a revision that we anticipate should not take you very long. 

Specifically, in response to Reviewer 2's comment 1, you show a western blot, but the reviewer specifically asked to see the expression in cortical sections, as its distribution within the tissue is the main interest here. In case the antibody is not good enough for IHC, please at the very least show a re-analysis of some existing scRNA-seq data to show which cell types Arl2 is most expressed in.

We will then assess your revised manuscript and your response to the reviewers' comments with our Academic Editor aiming to avoid further rounds of peer-review, although might need to consult with the reviewers, depending on the nature of the revisions.

**IMPORTANT - SUBMITTING YOUR REVISION**

*Resubmission Checklist*

*Published Peer Review*

*PLOS Data Policy*

*Blot and Gel Data Policy*

Sincerely,

Christian

Christian Schnell, PhD

Senior Editor

PLOS Biology

cschnell@plos.org

---

## [Editor Report · Decision Letter 3]

8 Jul 2024

Dear Dr Wang,

Thank you for your patience while we considered your revised manuscript "Arl2 Associates with Cdk5rap2 to Regulate Cortical Development via Microtubule Organization" for publication as a Research Article at PLOS Biology. This revised version of your manuscript has been evaluated by the PLOS Biology editors, the Academic Editor.

Based on our Academic Editor's assessment of your revision, we are likely to accept this manuscript for publication, provided you satisfactorily address the following data and other policy-related requests:

* We would like to suggest a different title to improve the accessibility: "Arl2 GTPase associates with the centrosomal protein Cdk5rap2 to regulate cortical development via microtubule organization"

* Please add the links to the funding agencies in the Financial Disclosure statement in the manuscript details

* Please state whether any other competing interests were present.

* Can you please add a statement to the corresponding figure legends where the raw data can be found? For example: "Source data can be found in S1 Data."

* DATA POLICY:

* CODE POLICY

We expect to receive your revised manuscript within two weeks. 

*Published Peer Review History*

*Press*

Sincerely,

Christian

Christian Schnell, PhD

Senior Editor

cschnell@plos.org

PLOS Biology

---

## [Editor Report · Decision Letter 4]

11 Jul 2024

Dear Dr Wang,

Thank you for the submission of your revised Research Article "Arl2 GTPase Associates with the Centrosomal Protein Cdk5rap2 to Regulate Cortical Development via Microtubule Organization" for publication in PLOS Biology. On behalf of my colleagues and the Academic Editor, Madeline Lancaster, I am pleased to say that we can in principle accept your manuscript for publication, provided you address any remaining formatting and reporting issues. These will be detailed in an email you should receive within 2-3 business days from our colleagues in the journal operations team; no action is required from you until then. Please note that we will not be able to formally accept your manuscript and schedule it for publication until you have completed any requested changes.

PRESS

Sincerely, 

Christian

Christian Schnell, PhD

Senior Editor

PLOS Biology

cschnell@plos.org